

# Implementing Microscopic Charcoal Particles Into a Global Aerosol-Climate Model

Anina Gilgen[1], Carole Adolf[2,3,*], Sandra O. Brugger[2,3,*], Luisa Ickes[1,4], Margit Schwikowski[5,3,6],
Jacqueline F. N. van Leeuwen[2,3], Willy Tinner[2,3,7], and Ulrike Lohmann[1]

[1]ETH Zürich, Institute for Atmospheric and Climate Science, Switzerland
[2]University of Bern, Institute of Plant Sciences, Switzerland
[3]University of Bern, Oeschger Centre for Climate Change Research, Switzerland
[4]Now at Stockholm University, Department of Meteorology, Sweden
[5]Paul Scherrer Institut, Villigen, Switzerland
[6]University of Bern, Department of Chemistry and Biochemistry, Switzerland
[7]ETH Zürich, Institute of Terrestrial Ecosystems, Switzerland
[*]These co-authors contributed equally to the paper and are considered joint second authors.

*Correspondence to:* Anina Gilgen (anina.gilgen@env.ethz.ch)

**Abstract.** Microscopic charcoal particles are fire-specific tracers, which are ubiquitous in natural archives such as lake sediments or ice cores. Thus, charcoal records from lake sediments are nowadays the primary source for reconstructing past fire activity. Microscopic charcoal particles are generated during forest and grassland fires and can be transported over large distances before being deposited into natural archives. In this paper, we implement microscopic charcoal particles into a global

aerosol-climate model to better understand the transport of charcoal on the large scale. Atmospheric transport as well as interactions with other aerosol particles, clouds, and radiation are explicitly simulated.

To estimate the emissions of the microscopic charcoal particles, we use recent European charcoal observations from lake sediments as a calibration dataset. We found that scaling black carbon fire emissions from the Global Fire Assimilation System (a satellite-based emission inventory) by a factor of $\approx 40$ matches the calibration dataset best. The charcoal validation dataset,

for which we collected charcoal observations from all over the globe, generally supports this scaling factor. In the validation dataset, we included charcoal particles from lake sediments, peats, and ice cores. The correlation coefficients for both the calibration and the validation dataset are positive and significantly different from zero, showing that the model captures a significant portion of the spatial variability. However, the model fails to reproduce the extreme spatial variability observed in the charcoal data. This can mainly be explained by the coarse spatial resolution of the model and uncertainties concerning fire

emissions. Furthermore, charcoal fluxes derived from ice core sites are much lower than the simulated fluxes, which can be explained by the location properties (high altitude and steep topography, which are not well represented in the model) of most of the investigated ice cores.

Global modelling of charcoal can improve our understanding of the representativeness of this fire proxy. Furthermore, it might allow to quantitatively validate past fire emissions provided by fire models. This might deepen our understanding of the

processes driving global fire activity.



# 1 Introduction

Fires are an important component of the Earth system and are closely linked to vegetation. They reduce biomass, influence the distribution of biomes, and alter biodiversity (Bond and Keeley, 2005; Convention of Biological Diversity, 2001). Furthermore, fires have a large impact on the atmosphere, mainly by emitting aerosol particles and greenhouse gases (Crutzen and Andreae, 1990) and to a smaller extent by altering the surface albedo (Gatebe et al., 2014). They threaten humans not only because of infrastructure and death risks but also because of carcinogenic smoke emissions (Stefanidou et al., 2008).

Biosphere, atmosphere, and humans are not only impacted by fires but also influence them: the occurrence and size of fires strongly depends on the vegetation properties (e.g. vegetation structure and moisture), on some climate variables (e.g. lightning frequency, precipitation, temperature), and on human behaviour (e.g. land use changes, fire fighting) (Hantson et al., 2016). In recent years, global fire models have become more advanced. Open questions still remain, e.g. regarding the complexity needed for global fire models (Hantson et al., 2016). Especially the anthropogenic influence on fires is difficult to simulate. Emissions of current fire models are calibrated to recent decades where satellite data gives valuable information on the occurrence of fires. Generally, a major goal of current research is to test the fire models against paleofire data: only if the models are able to reproduce past conditions, they may capture the key processes driving fires and provide trustworthy information about the future.

A number of natural archives provide information about paleo fires on different spatial and temporal scales. Sedimentary charcoal records from lakes and natural wetlands are unique because of the broad temporal and spatial coverage they provide, ranging from local to global spatial scales and decadal to millennial (Whitlock and Larsen, 2001). Recently, also charcoal particles originating from ice cores have been analysed (Isaksson et al., 2003). Beside charcoal particles, ice cores from glaciers and ice sheets also preserve other (potential) fire indicators such as black carbon (BC) or molecular fire tracers (Rubino et al., 2016). Due to their remote locations, ice cores can provide information on regional to subcontinental scale fire activity. Especially for the last $\approx 150$ years, ice cores generally have a sound chronology and a high temporal resolution, which allows to link recent ice core data directly to coinciding satellite observations or fire simulations. This is an advantage of ice cores compared to other charcoal fire records, which are undated in some cases and often have multi-decadal resolutions only (with some exceptions such as sediment traps or varved sediments).

Charcoal particles differ from BC (as defined in the aerosol community, see e.g. Bond et al., 2013) in terms of formation mechanism, size, density, and H-C and O-C ratios (Preston and Schmidt, 2006; Conedera et al., 2009). BC condenses as a secondary product from hot gases present in flames, thereby forming aggregates of small carbon spherules. Characteristic for BC particles are their submicron sizes and their very high carbon content, the latter resulting in pronounced absorption of visible light. In contrast, charcoal particles retain recognisable anatomic structures of their biomass source, cover the range from submicron to millimetre scale, and have considerably higher H-C and O-C ratios than BC (i.e. containing less carbon and thus absorbing less radiation). Both particles have in common that they are formed during biomass burning and are considered to be rather inert, unreactive substances.



Charcoal particles can be divided into microscopic ($D_\mathrm{M} > 10\,\mu\mathrm{m}$, where $D_\mathrm{M}$ is the maximum dimension of the particle) and macroscopic ($D_\mathrm{M} > 100\,\mu\mathrm{m}$) charcoal particles. In the most recent version of the Global Charcoal Database (GCDv3), more than thousand sites with charcoal data are collected (Marlon et al., 2016). However, comparing data from the GCD with output from fire models is challenging: the collected charcoal data is only comparable to a certain degree due to differences

in methods for extraction and counting, locations/environments, chronologies, particle sizes, values presented in percentages versus concentrations or influx, etc. To circumvent this problem, global synthesis studies such as Power et al. (2008) and Marlon et al. (2008) homogenised the variance of individual records with a Box-Cox transformation, rescaled the transformed data to the range (0, 1), and standardised it. The derived standardised scores (also called $Z$-scores) enhance the comparability of the data but give only information about the relative changes of charcoal deposition. However, absolute values of charcoal

fluxes (called influx or charcoal accumulation rate in the paleo science community) are crucial to validate fire models and fire emission inventories.

To link the location of charcoal emissions (i.e. fires) with the fluxes derived at the observation sites (e.g. lake sediment), the transport of the particles must be taken into account. Previous studies have already investigated the transport of charcoal particles, which can take place in either air or water depending on site conditions and record type (e.g. Clark, 1988a; Peters and

Higuera, 2007; Tinner et al., 2006a; Lynch et al., 2004). Instead of explicitly modelling the transport, many of these studies chose a statistical approach.

In his pioneering study, Clark (1988a) focused on the transport of charcoal particles in air. He expected that the transport in fire plumes (which uplift particles to high altitudes) is responsible for nearly the whole long-range transport of microscopic charcoal particles. Clark (1988a) calculated that the transport of charcoal particles can be subcontinental to global: although

charcoal particles are deposited relatively quickly due to their large sizes, their low density leads to considerably lower settling velocities compared to other supermicron particles (such as mineral dust).

More recently, Peters and Higuera (2007) and Higuera et al. (2007) used numerical models to simulate the major processes involved in macroscopic charcoal accumulation in lakes. Since they focused on charcoal particles from lake sediments, Higuera et al. (2007) considered not only fire conditions (size, location, and frequency) and transport but also sediment mixing and

sediment sampling of macroscopic charcoal, while microscopic charcoal remained unexplored.

In the very recent study of Adolf et al. (2017), a uniform European dataset of absolute charcoal fluxes is compared to satellite data of important fire regime parameters such as fire number, intensity, and burnt area at local and regional scales. Microscopic and macroscopic charcoal number fluxes are considered separately.

In this study, we explicitly simulate the aeolian transport and deposition of charcoal particles, which allows to quantitatively

compare simulated and observed charcoal fluxes. To model the transport of charcoal particles globally, we used the global aerosol-climate model ECHAM6-HAM2. We focus on microscopic charcoal particles, which primarily originate from fires in a radius of up to $100\,\mathrm{km}$ around the natural archive (Conedera et al., 2009), and are thus less influenced by specific site conditions (e.g. nearby burnable biomass). Part of microscopic charcoal particles can be transported over larger distances. For example, Hicks and Isaksson (2006) observed microscopic charcoal particles in Svalbard, which probably originated from the

neighbouring continents and thus had been transported for at least $\approx 1000\,\mathrm{km}$.



Using a global aerosol-climate model allows to calculate the meteorological conditions for the transport online. Furthermore, interactions of charcoal particles with other aerosol particles, with clouds, and with radiation can be considered. These factors might impact the removal processes of charcoal in the atmosphere and therefore where and when it is deposited.

Our main goals are to study the transport of microscopic charcoal particles on the global scale with a climate model and to
test the model performance using charcoal data from different paleo fire records. The structure of this paper is the following: we will first describe the charcoal data used for comparison with our simulations (Sect. 2). Subsequently, we describe the model including the implementation of charcoal particles as a new aerosol species into our aerosol scheme (Sect. 3). In the results and discussion section (Sect. 4), a comparison between model results and charcoal observations is shown as well as general atmospheric properties of the simulated charcoal particles such as lifetime and mixing state. In the conclusions (Sect. 5), we
summarise the key findings of this study.

## 2  Data

In this study, aerosol fire emissions (including charcoal) were prescribed using a satellite-based emission inventory. Since the emissions of microscopic charcoal were unknown, we estimated them by scaling the fire emissions of BC and comparing the model result to European charcoal observations (calibration dataset; Sect. 2.1). The derived scaling factor was then tested using
different charcoal observations from various regions around the globe (validation dataset; Sect. 2.2).

### 2.1  Data used for calibration

To calibrate our emissions, we used the data from Adolf et al. (2017). This dataset comprises charcoal observations from 37 lake sediments all over Europe (see Table A1). Compared to other parts of the world, biomass burning emissions from Europe are small. Nevertheless, we chose this dataset because of its uniqueness: i) annual fluxes are estimated very accurately owing to
the use of sediment traps, ii) due to the recent nature of the data (spring/summer 2012 to spring/summer 2015), it coincides with satellite-based fire emissions, iii) it includes a sufficiently large number of observation sites, iv) it covers a region sufficiently large to compare with a global model, and v) all charcoal samples were prepared with the same technique and all particles counted by the same person.

For nearly all sediments considered in this study, we can assume that the transport of charcoal takes place predominantely
in air, not in water. However, for one lake in Southern Spain (Laguna Zóñar) and one in Switzerland (Mont d'Orge), surface runoff is expected to be important because of the bare soil around the lakes. Surface runoff can transport deposited charcoal particles from the soil to the lake and thus enhance the number of charcoal particles in the sediment traps. Therefore, data from these two sites must be interpreted with caution.

Charcoal particles were counted in pollen slides with a magnification of 200-250×. Samples for microscopic charcoal
analysis were treated following palynological standard procedures (Stockmarr, 1971; Moore et al., 1991). All black, completely opaque, and angular particles (Clark, 1988b) with a minimum $D_M$ of $10\,\mu m$ and a maximum $D_M$ of $500\,\mu m$ were counted following Tinner and Hu (2003) and Finsinger and Tinner (2005).



## 2.2 Data used for validation

For validating the model results, we used microscopic charcoal observations covering different parts of the world, which are independent of the calibration data set. Table A2 summarises the locations and the time (period) of the observations used for validation. Overall, data from 32 lake sediments and peats was compiled using the Alpine Pollen Database of University of Bern (ALPADABA). While many charcoal observations from lake sediments and peats exist, charcoal particles have so far only been studied in a handful of ice cores (e.g. this study; Isaksson et al., 2003; Eichler et al., 2011; Reese et al., 2013). In our analysis, we include five ice core records. Three of them were obtained in the frame of the project "Paleo fires from high-alpine ice cores" (which also includes this study), in which also charcoal data from the Eurocore 89, Greenland, was analysed. One of them is from Belukha glacier, Siberian Altai (Eichler et al., 2011).

The selected data is as homogeneous as possible: to compare the dataset with our simulated results, only number fluxes of charcoal particles with a lower threshold of $D_{\mathrm{M}} = 10\,\mu\mathrm{m}$ were considered. We excluded data using a different threshold or reporting no information from which we could calculate fluxes. Since the preparation technique also influences the estimated fluxes (Tinner and Hu, 2003), we furthermore ensured that the sample preparation and charcoal identification for the validation dataset are identical to that of the calibration dataset. The only exception is the data from Connor (2011). Instead of counting the number of charcoal particles above $10\,\mu\mathrm{m}$, Connor (2011) measured the charcoal area following the method from Clark (1982). To compare it with the simulated number fluxes, the linear regression from Tinner and Hu (2003) for Lago di Origlio was applied to the observed data to convert charcoal area to number.

For the lake sediments and peats, we include additional information about the dating of the records in Table A2. The sediment age was used to calculate sediment accumulation rates. Based on the original chronologies, we assumed a linear sediment accumulation between the two youngest charcoal samples to calculate a sediment accumulation rate from which we then derived the charcoal flux for the uppermost sample of the record. By assuming a linear sediment accumulation, we may underestimate true values given that surface sediments are not compacted yet. The surface of the sediment core usually reflects the time of drilling. Therefore, the older the youngest dated point of the core, the larger the uncertainty of the most recent sediment accumulation rate and, consequently, the charcoal fluxes. Furthermore, the uncertainty of the fluxes depends on the dated material and the dating method (both listed in Table A2).

The ice cores considered for validation derive from Colle Gnifetti (Switzerland), Tsambagarav (Mongolia), Belukha (Russia), Illimani (Bolivia), and Summit (Greenland), thus spanning a wide range of the globe. An exotic *Lycopodium* spore marker was added to the melted samples, which were then evaporated to reduce the volume and afterwards treated in the same manner as the standard sediment samples.

We only take into account data that is more recent than 1980 in the validation dataset as we had to find a compromise between including observations reflecting the fire conditions of the simulated period (2005-2014) and observations coming from many different locations, but which are older than the simulation period.




## 3 Methodology

### 3.1 Modelling charcoal particles in ECHAM6-HAM2

ECHAM6-HAM2 is a global climate model (ECHAM) coupled with an aerosol model (HAM) and a 2-moment cloud microphysical scheme. For more information about the model, we refer to Stier et al. (2005), Lohmann et al. (2007), Zhang et al. (2012), Stevens et al. (2013), and Neubauer et al. (2014). Since this is the first time that microscopic charcoal has been implemented into a global aerosol-climate model, we will in the following thoroughly describe which aspects need to be considered. First, we will describe general physical properties of microscopic charcoal and how these are represented by the model (Sect. 3.1.1). Second, we will describe how a lifecycle of charcoal particles is simulated, i.e. from the emissions (Sect. 3.1.2) via atmospheric interactions (Sect. 3.1.3, 3.1.4) through to deposition (Sect. 3.1.5). In the end, some diagnostics complementary to the existing model output will be mentioned briefly (Sect. 3.1.6).

### 3.1.1 Size distribution, shape, and density

HAM uses the so-called M7 scheme (Vignati et al., 2004), which distinguishes seven aerosol modes classified by their size and solubility: soluble nucleation mode (number geometric mean radius $r_g < 5\,\text{nm}$), soluble Aitken mode ($5\,\text{nm} < r_g < 50\,\text{nm}$), insoluble Aitken mode, soluble accumulation mode ($50\,\text{nm} < r_g < 500\,\text{nm}$), insoluble accumulation mode, soluble coarse mode ($500\,\text{nm} < r_g$), and insoluble coarse mode. Each of these modes is log-normally distributed, and the total aerosol particle size distribution is described by a superposition of the seven modes. To implement charcoal particles, we extended the scheme by two additional modes (M9 scheme), namely by a soluble giant and an insoluble giant mode. We restricted neither the upper nor the lower bound of the giant mode but the $r_g$ of the emitted (i.e. initial) size distributions was set between $0.5$ and $5\,\mu\text{m}$ (see Sect. 3.1.2). When a particle size distribution grows in M7, part of its mass and number is shifted to the next larger mode, e.g. from the nucleation to the Aitken mode. To simplify diagnostics, we did not allow shifts from the coarse to the giant mode.

In HAM, all aerosol particles are assumed to be spherical. This condition is not fulfilled for charcoal particles but at least microscopic charcoal particles seem to have a shape closer to a sphere than macroscopic charcoal particles (Crawford and Belcher, 2014). To compare our result with observations, we therefore use the volume-equivalent radius ($r_{eq}$) of charcoal particles. To estimate $r_{eq}$, the geometry of charcoal particles must be considered. Some studies analysed the shape of charcoal particles and reported their aspect ratios $R = \frac{D_M}{D_m}$, where $D_m$ is the minimum dimension of a particle. We briefly summarise the findings concerning $R$ in literature before explaining which range of $R$ we consider in our model simulations.

The right-skewed histogram by Clark and Hussey (1996) shows a distinct maximum in the bin $R = 1.5\text{-}2$, and the mean aspect ratio is $2.36 \pm 1.53$. While Clark and Hussey (1996) used 9 sites in temperate eastern North America for their analysis, Tinner and Hu (2003) studied charcoal particles from different biomes, namely Lago di Origlio (Switzerland; warm-temperate chestnut forest), Grizzly Lake (Alaska; spruce forest), and Wien Lake (Alaska; shrub birch tundra, poplar forest, and boreal forest). For the three sites, they report aspect ratios of $R = 1.9$, $R = 1.7$, and $R = 1.6$, respectively. Crawford and Belcher (2014) measured the aspect ratios of both microscopic and macroscopic charcoal particles. For microscopic charcoal ($D_M$ up to $100\,\mu\text{m}$), they found aspect ratios of 1.8 and 2.4 for charcoal from wood and grass, respectively. It is worth mentioning that





they used a cross-sectional area of $315\,\mu m^2$ as the lower threshold, which corresponds to a $D_M$ of about $11.5 - 13.4\,\mu m$ for wood and $13.2 - 15.5\,\mu m$ for grass (assuming rectangular/elliptical cross-sections), i.e. a slightly larger $D_M$ than the threshold of $10\,\mu m$ used in this study.

All of these measurements of $R$ lie in the same range. For our study, we chose $R = 2$ as an initial estimate. The third, non-visible dimension of the particle is expected to be smaller or equal to $D_m$ for particles detected in pollen slides since the particles may tend to lie flat on the slides (Clark and Hussey, 1996). For simplicity, we describe the shape of the charcoal particles with a rectangular cuboid (see Clark and Hussey, 1996). Assuming that the non-visible axis equals the minor axis $D_m$ (which is rather an upper estimate), the equivalent-volume radius $r_{eq}$ is given by:

$$V_{cuboid} = V_{sphere} \tag{1}$$

$$D_M \cdot \frac{D_M}{R} \cdot \frac{D_M}{R} = \frac{4}{3} \cdot \pi \cdot r_{eq}^3 \tag{2}$$

$$\rightarrow r_{eq} \approx 0.62 \cdot \frac{D_M}{R^{\frac{2}{3}}}, \tag{3}$$

where $V$ stands for volume. The typical lower threshold for microscopic charcoal particles is $D_M = 10\,\mu m$, which corresponds to an equivalent-volume radius of $r_{eq} \approx 3.9\,\mu m$. However, since the aspect ratio tends to increase with charcoal size (Crawford and Belcher, 2014), $R$ of the lower threshold ($D_m = 10\,\mu m$) might be smaller than the mean or median $R$ for $D_m > 10\,\mu m$. In the model, we cannot account for a size-dependent $R$. For this study, it is important that the lower threshold of the counted and simulated charcoal particles match well since these small particles have higher number concentrations than larger particles (Clark and Hussey, 1996; Tinner et al., 1998). As a lower estimate for our test simulations (see Sect. 3.2), we therefore use $R = 1.33$, which corresponds to the often applied, observation-based threshold of $75\,\mu m^2$ for microscopic charcoal cross-sections (e.g. Tinner et al., 2006b) and which results in $r_{eq} = 4.9\,\mu m$. Based on the before mentioned observations from Clark and Hussey (1996) and Crawford and Belcher (2014), $R = 2.4$ is considered to be an upper bound.

A distinct characteristic of charcoal particles is their low density. Renfrew (1973) reports values of $0.3$-$0.6\,g\,cm^{-3}$, Sander and Gee (1990) similar values of $0.45$-$0.75\,g\,cm^{-3}$. Hence, we chose a particle density of $0.5\,g\,cm^{-3}$ as an initial guess, which lies in the middle of these ranges. For the test simulations, we considered values where both observations overlap, i.e. from $0.45$ to $0.6\,g\,cm^{-3}$ .

## 3.1.2 Charcoal emissions

Thanks to fire emission inventories based on satellite data, we have a good knowledge about where and when fires of which sizes occurred in the last 1-2 decades. Nevertheless, aerosol emissions from fires are still uncertain. This is caused to a large degree by the pronounced variability of fires: emission factors (which relate the mass of the burnt vegetation to the mass of emitted aerosol particles) vary considerably depending for instance on vegetation type, fire temperature, or fire dynamics. To our knowledge, no study has estimated the emission factors of microscopic charcoal particles so far. Clark et al. (1998) and



Lynch et al. (2004) focused on macroscopic charcoal when estimating mass emission fluxes, therefore these values are not comparable.

Airborne measurements of aerosol particles from fires usually have upper cutoff sizes of a few micrometres or less (e.g. Johnson et al., 2008; May et al., 2014). The aircraft measurements by Radke et al. (1990) are exceptional since they include

particles with sizes up to $3\,\mathrm{mm}$, therefore covering the whole size range of charcoal. In their study, they set three fires in North America. The measured particle size distribution showed similar shapes for all of these burns. Radke et al. (1990) report that a considerable fraction of the particles measured in the plumes were larger than $45\,\mathrm{\mu m}$ in diameter. From their data, we estimate that the mass emission fluxes of microscopic charcoal particles should be on the same order of magnitude as the mass emission fluxes of submicron particles, which is usually dominated by organic carbon (OC) in fire plumes (Desservettaz et al., 2017).

By that we assume that all of these large particles are indeed charcoal and not ash or other large particles emitted from fires.

Since both BC and charcoal particles form under conditions when oxygen is limited in the burning process, we decided to scale BC mass emissions from fires to derive charcoal mass emissions. As a starting point for the scaling factor, we assume that the mass emission fluxes of microscopic charcoal are comparable to those of submicron particles and thus arrive at a factor of $\approx 10$ based on the ratios of BC to total submicron particles and to OC (Desservettaz et al., 2017; Akagi et al., 2011; Sinha et al.,

2003). Note that a larger factor of $\approx 34$ might also be realistic since scaling aerosol emissions from GFAS by a factor of 3.4 leads to a better agreement between simulated and observed aerosol optical depth for both the global Monitoring Atmospheric Composition and Change (MACC) aerosol system and ECHAM6-HAM2 (Kaiser et al., 2012; von Hardenberg et al., 2012). Then we adjust this scaling factor until the simulated charcoal fluxes agree with the calibration dataset (Sect. 2.1).

To describe the fire emissions, we use BC, OC, and $SO_2$ mass emissions at a 3-hourly resolution by combining the daily

emissions from the Global Fire Assimilation System (GFASv1.0 until September 2014, GFASv1.2 afterwards) with the daily cycle from the Global Fire Emissions Database (GFED; Kaiser et al., 2012; Mu et al., 2011). GFAS emissions are based on fire radiative power and make use of vegetation-specific aerosol emission factors following Andreae and Merlet (2001, with annual updates by M. O. Andreae). The strongest spurious signals originating from industrial activity, gas flaring, and volcanoes should be masked. However, in our simulations we found unrealistically high charcoal emissions over Iceland. These "emissions" are

most likely caused by lava, which emits a signal at the same wavelength at which fires are detected. As an example, the volcano Bardarbunga caused huge eruptions over Iceland in August/September 2014, coinciding with extremely high fire emissions in GFAS ($2.32 \cdot 10^{-11}\,\mathrm{kg\,m^{-2}\,s^{-1}}$ averaged between $62°\,\mathrm{N}\ 26°\,\mathrm{W}$ and $67°\,\mathrm{N}\ 11°\,\mathrm{W}$ for September compared to global mean emissions of $1.57 \cdot 10^{-13}\,\mathrm{kg\,m^{-2}\,s^{-1}}$ for the same month). Therefore, we decided to mask all fire emissions over Iceland for our simulations. Furthermore, note that some fires are not detected by the satellite when clouds obscure the fire radiative power

signal or when the signal is below the detection limit (which depends on the distance to sub-satellite track; Kaiser et al., 2012). Other uncertainties of biomass burning emissions include for example uncertainties in emission factors or land cover maps (Akagi et al., 2011; Fritz and See, 2008). More details about GFAS can be found in Kaiser et al. (2012).

Observations show that the larger the microscopic charcoal particles, the smaller their corresponding number concentration (e.g. Clark and Hussey, 1996). This implies that the number geometric mean radius $r_\mathrm{g}$ of our emitted charcoal size distribution

should be smaller than the lower threshold of microscopic charcoal detection ($D_\mathrm{M} = 10\,\mathrm{\mu m}$), i.e. the observations lie on the





descending branch of the emitted log-normal size distribution (see Fig. A2). For deposited charcoal number size distributions, Clark and Patterson (1997) suggest that the median radius might indeed be somewhat smaller. From their data, we estimate that the peak in the number size distributions lies around $D_\mathrm{M} = 3.15\,\mu\mathrm{m}$, which corresponds to $r_\mathrm{eq} = 1.2\,\mu\mathrm{m}$ using $R = 2$. Of course, the size distribution of particles in air and the deposited size distribution could differ considerably. The further away from the fire, the smaller we expect the number geometric mean radius to be because larger particles are deposited more quickly. As a consequence, the number geometric mean radius of the samples that Clark and Patterson (1997) analysed should be larger than $r_\mathrm{eq} = 1.2\,\mu\mathrm{m}$ upon emission. The airborne measurements by Radke et al. (1990) only show one clear maximum in the number size distribution at radius $r = 0.05\,\mu\mathrm{m}$, which we attribute to other aerosol particles than charcoal (e.g. BC and OC). There is however a distinct flattening of the negative slope above $r = 0.5\,\mu\mathrm{m}$, which could well be caused by an increase in the charcoal particle number concentration. Based on these two studies, we estimate that the number geometric mean radius at emission lies in the range between $0.5$ and $5\,\mu\mathrm{m}$.

In contrast to the studies by Clark (1988a) and Higuera et al. (2007), our fire plume heights depend on the planetary boundary layer (PBL) height (Veira et al., 2015), which is illustrated in Fig. A1. If the PBL height is lower than $4\,\mathrm{km}$, $75\,\%$ of the fire emissions are distributed between the surface and the model layer below the PBL height (at a constant mass mixing ratio), $17\,\%$ are injected in the first model layer above the PBL height, and $8\,\%$ in the second layer above the PBL height. In the rare cases when the PBL height is larger than $4\,\mathrm{km}$, the plume height is set to the PBL height and the emissions are equally distributed from the surface to the model layer below the PBL height.

We assume that all charcoal is emitted as insoluble particles because of their rather high carbon content and inertness (Preston and Schmidt, 2006). Observations have shown that BC, which is like charcoal an inert and unreactive substance, can take up soluble material and undergo further hygroscopic growth (Shiraiwa et al., 2007; Zhang et al., 2008). Hence, we assume that the same holds for charcoal particles, i.e. that charcoal particles can become soluble and thus be shifted to the soluble mode. This is explained in the following section.

### 3.1.3 Interactions with other aerosol particles

Charcoal particles can be shifted from the insoluble giant to the soluble giant mode by two processes: i) by Brownian coagulation with soluble particles from the nucleation or Aitken mode, and ii) by condensation of sulphuric acid on the particle surface. Coagulation with larger modes is not considered because the Brownian motion of these particles is very low and coagulation is therefore not effective. Schutgens and Stier (2014) reported that even coagulation between the Aitken (BC, OC, sulphate) and the coarse mode (dust) is negligible, which suggests that the same might be the case for the Aitken and the giant mode (charcoal). Since charcoal particles – in contrast to dust – are co-emitted with BC, OC, and sulphate, we decided to nevertheless implement the coagulation between the giant and the Aitken mode. By coagulation, the aerosol species BC, OC, and sulphate can be transferred to the soluble giant mode. The soluble giant mode is therefore a mixture of different aerosol species, whereas the insoluble giant mode is exclusively comprised of charcoal.





In our model, the condensation of sulphate shifts the charcoal particle to the soluble mode when at least one mono-layer of sulphate covers the surface of the charcoal particle. Therefore, large charcoal particles are less likely to be transferred to the soluble mode by condensation of sulphate than small charcoal particles.

It is assumed that the soluble giant mode is *internally* mixed, i.e. each individual aerosol particle consists of all components present in the mode. As soon as charcoal has been shifted to the soluble mode, the particles can grow further by water uptake when hygroscopic material like sulphate is present. In-cloud produced sulphate mass can sometimes be added to the giant soluble mode when cloud droplets evaporate (see Appendix A).

### 3.1.4 Interactions with microphysics and radiation

In the soluble giant mode, mixed aerosol particles containing charcoal can act as cloud condensation nuclei following the Abdul-Razzak and Ghan (2000) activation scheme. Charcoal itself does not dissociate.

Whether charcoal particles can initiate freezing or not influences their atmospheric lifetime. This is because ice formation in a cloud, followed by the Wegener-Bergeron-Findeisen process, usually leads to precipitation and therefore to wet deposition of charcoal particles. Theoretically, we think that charcoal might be an ice nucleating particle (INP) since it fulfills the criteria found for INP: it does not dissociate, it is large, and it has a complex surface structure with cracks in which ice embryos might form more easily (Hoose and Möhler, 2012; Lohmann et al., 2016). However, to our knowledge nobody has measured the ice nucleating ability of charcoal particles so far. Due to the lack of measurements and freezing parameterisations, we therefore assume that charcoal particles cannot initiate freezing of cloud droplets.

Aerosol particles also interact with radiation. The atmospheric lifetime of charcoal particles could be enhanced by light absorption leading to lofting. In ECHAM6-HAM2, the refractive index (RI) of the aerosol species must be defined at different wavelengths. The radiative properties of BC or other small, light-absorbing aerosol particles have widely been studied, and the radiative properties of coals with different carbon contents have also been measured. However, nobody has measured the RI of microscopic charcoal particles as far as we know. We used the same RI as for dust; for explanation, see Appendix B.

### 3.1.5 Removal processes

Aerosol particles can be removed by three processes in HAM: by wet deposition, graviational settling, and dry deposition. Wet deposition in ECHAM6-HAM2 includes both in-cloud and below-cloud scavenging (Croft et al., 2009, 2010). Furthermore, the calculation distinguishes between liquid, mixed-phase, and ice clouds, as well as between stratiform and convective clouds. The wet deposition calculation explicitly considers the sizes and the solubility of the aerosol particles. To prevent numerical instability, settling aerosol particles cannot fall through more than one model layer within one timestep. However, this should not considerably change the spatial gravitational settling pattern (for details, see Appendix C). In contrast to gravitational settling and wet deposition, dry deposition is only calculated near the surface. It accounts for the fact that a higher surface roughness leads to an increased aerosol flux to the surface because of turbulence. The surface roughness varies for different surface types, e.g. forest, water, or ice. Since gravitational settling is artifically slowed down near the surface (Appendix C), dry deposition might take over and could therefore be somewhat overestimated.



### 3.1.6 Additional diagnostics

As mentioned in Sect. 3.1.2, we estimate the number geometric mean radius of the emitted charcoal size distribution to lie in the range 0.5-5 µm. This implies that a substantial contribution of the simulated charcoal particles are smaller than $D_{\mathrm{m}} = 10\,\mu\mathrm{m}$ and are therefore not included in the counts under the microscope. When comparing the simulated number fluxes to the surface with observations, we therefore want to exclude these small particles in our diagnostics. However, in the standard setup of ECHAM6-HAM2, only the total surface fluxes for each giant mode are calculated. To circumvent this problem, we implemented additional diagnostics which calculate how many particles above a threshold radius are deposited. More information can be found in Appendix D.

### 3.2 Model simulations

In this study, we used a model resolution of T63L31, which corresponds to a gridbox size of $1.9° \times 1.9°$ ($\approx 200\,\mathrm{km} \times 200\,\mathrm{km}$ at the equator) with 31 vertical layers. For all simulations, we used a spin-up time of three months. First, we conducted test simulations to find suitable values for charcoal emission factors and some uncertain parameters. These test simulations were nudged towards 6-hourly ERA-Interim data from April 2012 to May 2015 to cover the same time period as the calibration dataset used to evaluate the model performance. As mentioned previously, we increased the BC mass emissions by a scaling factor ($SF$) to estimate the charcoal emissions. Three charcoal parameters were varied in the test simulations: the threshold radius (above which charcoal particles are counted), the emission number geometric mean radius, and the density. As an initial guess, we set the emission number geometric mean radius to $r_{\mathrm{eq}} = 2.5\,\mu\mathrm{m}$, the threshold radius to $r_{\mathrm{eq}} = 3.9\,\mu\mathrm{m}$ (corresponding to $R = 2$), and the density to $0.5\,\mathrm{g\,cm}^{-3}$. In Table 1, we refer to this simulation as *remi2.5,rthr3.9,dens0.5*. The values were varied in the ranges derived from literature (see Sect. 3.1.2).

Finally, we conducted a nudged and a free simulation of 10 years each (January 2005 to December 2014) with the derived emissions and compared our results with the observations described in Sect. 2.2 and 3.1.1.

For all simulations, we used 3-hourly fire emissions based on daily GFAS emissions (see Sect. 3.1.2). The other prescribed aerosol emissions are monthly means and do not show interannual variability. For most of these aerosol particles, we used present-day emissions (year 2000) from ACCMIP (Lamarque et al., 2010). Dust, sea salt, and oceanic dimethyl sulphide emissions were calculated online.

## 4 Results and discussion

### 4.1 Calibration of emissions

We conducted test simulations and compared the result to the European observations from Adolf et al. (2017). Table 1 shows some parameter combinations which yield Pearson correlation coefficients larger than 0.2. In all test simulations, the Spearman rank correlation is much higher than the Pearson correlation. The main reason for that are some observations with clearly larger charcoal fluxes than the simulated values ("outliers"), which can nicely be seen in Figure 1. Two of these outliers



(black in Fig. 1) are sites expected to be influenced by surface runoff (Laguna Zóñar and Mont d'Orge), which explains the discrepancy between observations and model results. Removing these two points from the calculation causes the Pearson correlation to increase (e.g. from 0.27 to 0.33 for simulation *remi2.5,rthr3.9,dens0.5*). The other outliers (dots in Fig. 1 above $40000 \, \# \, \mathrm{cm}^{-2} \, \mathrm{y}^{-1}$) are sites located in Sicily and Southern France. A minor part of the deviation might be due to the proximity

of these sites to the ocean. In this case, the gridboxes contain both land and ocean, which leads to an underestimation of charcoal emission fluxes over land in the model.

In all test simulations, the variability of charcoal fluxes is underestimated. The last column in Table 1 shows the quartile coefficient of dispersion, which is a normalised and robust variability measure ($\frac{Q_3 - Q_1}{Q_3 + Q_1}$, where $Q_1$ and $Q_3$ are the first and third quartile, respectively). The simulation with the highest variability (*remi5,rthr4.9,dens0.6*) has a comparably low correlation.

As best parameters, we therefore choose *remi2.5,rthr4.9,dens0.6*: together with *remi2.5,rthr4.9,dens0.5*, these parameters have the highest Pearson correlation of all simulations. In addition, the variability is somewhat larger in *remi2.5,rthr4.9,dens0.6* than in *remi2.5,rthr4.9,dens0.5*. Since it is not feasible to conduct simulations with all possible parameter combinations within their realistic ranges, we admit that other parameter combinations could describe the observational data similarily well.

For the chosen parameters, we found that $SF$ on the order of 40 is in best accordance with the observations (see Fig. 2;

roughly the same number of observations lays above and below the 1:1 line). Since the simulated variability is too low, we underestimate the charcoal fluxes e.g. over Sicily and overestimate them e.g. over Scandinavia (see Fig. 3). We think that the two following reasons are mainly responsible for the larger variability in the observations compared to the model:

- Model resolution: sub-grid variability cannot be resolved by the model, i.e. the simulated emissions and depositions are an average over the whole gridbox. In contrast, the observation sites can differ by large amount e.g. concerning the

distance to burnable biomass, especially in the highly fragmented landscapes of Europe.

- Uncertainties in fire emissions: some fires might not be detected by the satellite (e.g. due to dense clouds) and therefore not be accounted for in the simulated emissions. Furthermore, charcoal particle emissions could show a different variability concerning vegetation than BC does, i.e. the charcoal emissions per mass of burnt biomass might vary more between different vegetation types than we assumed.

Although the model underestimates the variability, it is able to capture the European South-North decrease in charcoal fluxes (Fig. 3). Furthermore, the correlation is significant (at the $5\%$ significance level) when we remove the two sites that are expected to be influenced by surface runoff: 0.39 and 0.75 for the Pearson and the Spearman rank correlation, respectively. In the next section, we will validate the model with observations from different regions of the world.

## 4.2 Comparison with observations

In this section, we compare our simulated charcoal fluxes with independent observations. Here, we show results for the nudged 10-year model simulation; those of the free 10-year simulation are very similar (for comparison, Fig. A4 shows the same as Fig. 4 but for the free simulation). For the three ice cores spanning a multi-annual recent period, we average the model output





over the same time periods (2005 to summer 2009 for Tsambagarav, 2005 to 2014 for Colle Gnifetti, and 2008 to 2014 for Illimani). For all other observations, we use the mean over the whole simulation for comparison.

As for the calibration simulations described in Sect. 4.1, the high variability in the observations is not reproduced by the model (see Fig. 4). For Bhutan, Italy, Switzerland, and Georgia, several lake sediment samples were collected in a small
geographical region and are therefore not further distinguished in Fig. 4 (black, yellow, green, and orange symbols, respectively; medians over these samples shown as large pentagrams). The regional medians of the observations are rather close to the simulated median charcoal fluxes, indicating that the simulated fluxes are representative for a large scale. While the simulated median over Italy is somewhat underestimated, it is overestimated for Switzerland and Bhutan. For Georgia, the simulated and the observed median compare very well. Note that most of the data is also shown on a linear scale (Fig. 5), where we zoom in
for better visibility (red frame in Fig. 4). The data from Connor (2011) (orange crosses) is the only one originally measured in area fluxes and afterwards converted to number fluxes (see Sect. 2.2). The converted number fluxes compare well with the other observations and the model results, which indicates that the regression from Tinner and Hu (2003) can indeed be applied in this case.

Most simulated fluxes deviate less than one order of magnitude from the observations, providing evidence that the simulated
results are in good agreement with observed values at the sites. However, the charcoal flux values are highly overestimated for all ice cores (triangles), for three peats in the alpine region (Mauntschas, Rosaninsee, and Wengerkopf), and for the sediment from Lake Kharinei (northern Russia). The model probably overestimates the fluxes at the ice core sites because of their high location within complex topography. The model is not able to simulate these high locations correctly since the surface altitude is constant over the whole gridbox, i.e. topography is "averaged out". The simulated gridbox averages are therefore
not comparable to the ice core measurements. In reality, ice cores are located above the top plume height of most fires (Rémy et al., 2017), which may prevent transport of charcoal particles to them. Furthermore, the simulated fire emission height has a bias towards higher plume heights (see Sect. 4.1), which likely also contributes to the overestimation of simulated charcoal fluxes for sites above $4000\,\mathrm{m}$ (Rodophu-2 and all ice cores except Greenland). In addition, we expect that this bias leads to an overestimation of the simulated transport of charcoal to remote locations, which could explain the high simulated fluxes
at Lake Kharinei and Greenland. Another explanation for the overestimated simulated fluxes in Greenland is an increase in fire activity: GFAS data between 2003 and 2015 suggests that the fire emissions in Greenland might have increased in the last years. Fire activity was recorded in the years 2003, 2007, and all years since 2011, with highest aerosol emissions occurring in 2015. Therefore, it is possible that the fire activity was lower in 1989 (when the ice core was drilled) than in the simulated period. For the alpine sites, the observed fluxes might again not be representative for the whole gridbox due to the small-scale,
heterogeneous landscape around these observation sites (fire emissions/vegetation cover are constant in one model gridbox).

Overall the chosen scaling factor ($SF = 40$) describes the data well, i.e. a *global* charcoal scaling factor seems to be justified. However, the validation dataset does not cover certain regions (e.g. Africa or Australia) and is biased towards northern mid-latitudes. The correlation between observed and simulated fluxes is 0.58 and 0.51 for the Pearson and the Spearman rank correlation, respectively, and in both cases statistically significant.



### 4.3 Global distribution considering all microscopic charcoal particles

The global microscopic charcoal burden, i.e. the vertically integrated mass of all charcoal particles in the atmosphere (not only those above the threshold radius), is shown in Figure 6a averaged over the 10-year nudged simulation. Note that the larger the particles, the larger their mass and thus their contribution to the burden. As expected, the burden is highest where most biomass

burning emissions occur, namely in the tropics followed by the high northern latitudes (mainly Siberia, North America; Kaiser et al., 2012). The simulated global mean burden is $7.97 \times 10^{-7} \, \mathrm{kg \, m^{-2}}$, i.e. approximately three times larger than the burden of BC, which is $2.64 \times 10^{-7} \, \mathrm{kg \, m^{-2}}$ including all BC sources (shown in Fig. 6b). Although the largest charcoal burdens occur near the emission sources, significant fractions of charcoal mass are transported hundreds of kilometres in the model, which is for example the case near the east coast of North America or the west coast of Central Africa. However, the burden gives no

indication about the size of the particles, i.e. a large part of the far-transported particles are likely to be submicron particles.

Most of the charcoal resides in the insoluble mode in terms of number and mass and only a small fraction ($\approx 10\%$) is shifted to the soluble mode (see Fig. 7). The small contribution of the soluble mode can be explained by the large size of the charcoal particles (limiting amount of coating material) and the related short atmospheric lifetime. Beside charcoal, the soluble mode is predominantely comprised of sulphate (and water), while the mass contributions of BC and OC are small (not shown).

### 4.4 Deposition of microscopic charcoal particles

As expected, the different atmospheric removal processes for charcoal particles above the threshold radius differ in geographic distribution (see Fig. 8). While gravitational settling and dry deposition become less important the larger the distance to the emission source, this is not generally the case for wet deposition. Large wet deposition fluxes are observed where (simulated) precipitation is high, e.g. along the Atlantic storm track. Contrary to gravitational settling, dry deposition depends on the

surface properties (Stier et al., 2005). Therefore, dry deposition fluxes are small over the ocean compared to over land.

Overall, gravitational settling is the most important removal process, followed by dry deposition and then wet deposition. Although gravitational settling and dry deposition dominate the global charcoal deposition, wet deposition is the dominant removal process in some remote regions like northern Scandinavia.

### 5 Conclusions

Charcoal records from lake sediments are widely used to reconstruct past fire activity. More recently, charcoal particles have also been studied in ice cores. In this paper, we implemented microscopic charcoal particles into a global aerosol-climate model. Comparing simulated with observed charcoal fluxes might help to quantitatively reconstruct past fire activity. A recent and comprehensive charcoal dataset from Europe was used for calibration of model emissions. Increasing BC fire emissions by a factor of 40 resulted in the best match between the model and observations. Although the model is not able to reproduce the high

local variability of the observations, the correlation is positive and statistically significant, indicating that the model captures the large-scale pattern of charcoal deposition reasonably well. The charcoal fluxes for the validation dataset, which covers




different locations across the globe, are well captured with the constant charcoal scaling factor derived from the European calibration dataset. However, our validation dataset consists mostly of samples from Northern mid-latitudes. We also found an underestimation in variability for the validation dataset but a positive, statistically significant correlation between modelled and observed fluxes. The model shows a systematic positive bias for the ice core observations, which is likely due to the high
alititude of the ice core sites as well as the complex topography around them.

As expected, the largest simulated charcoal deposition fluxes occur near fires. However, the model suggests that a non-negligle amount of microscopic charcoal particles is transported over large distances and therefore reaches remote locations (although comparisons with observations indicate that the model might overestimate long-range transport). In the model, only a small fraction of charcoal particles ($\approx 10\,\%$) is mixed with soluble material in the atmosphere.
More systematic and standardised observations of microscopic charcoal (as number fluxes with maximum particle dimension $> 10\,\mu m$) could help to improve the comparison and to verify that the scaling factor proposed here ($SF = 40$) indeed describes the data well on the global scale. In future studies, our new framework allows global modelling of charcoal and other biomass burning relevant tracers such as black carbon, which may improve the understanding of the representativeness of individual fire proxies. In addition, simulating microscopic charcoal particles using the found scaling factor might allow us to quantitatively
validate past fire emissions provided by fire models. The validation of fire models is essential to improve the understanding of the key drivers of fires and to gain confidence in projections of future fire activity.

## Appendix A: In-cloud produced sulphate

Sulphate aerosols can be produced in cloud droplets when $SO_2$ reacts with $O_3$ or $H_2O_2$. When cloud droplets evaporate, aerosol particles remain, the size of which depends on the mass and chemistry of the foreign material in the cloud droplets
(Mitra et al., 1992). The reaction with $H_2O_2$ is considered to be the dominant pathway (Seinfeld and Pandis, 2006). Since the $H_2O_2$ concentration is often the limiting factor for the reaction with $SO_2$, most of sulphate is added to those particles activated early in the cloud, i.e. the best cloud condensation nuclei (Harris et al., 2014). Therefore, we distribute the sulphate mass produced in-cloud among the larger soluble aerosol modes (accumulation, coarse, and giant) in case these modes exist. If none of the three modes exist, a new soluble coarse mode is created.

## Appendix B: Radiative index of charcoal

Many studies (e.g. Habib and Vervisch, 1988) report that higher H-C ratios result in a smaller imaginary part of RI, i.e. in a smaller absorption component. However, Bond and Bergstrom (2006) reviewed the radiative properties of different carbon-containing substances with focus on light-absorbing aerosol particles and found that the number of $sp^2$ bonds (more precisely: the extent of $sp^2$-islands) matters most. More $sp^2$-islands result in higher absorption because $sp^2$-bonded carbon is arranged
in planar layers, which allows the $\pi$-electrons to move freely. Although the light absorption is closely related to the imaginary



part of RI, it is important to note that absorption also impacts the real part. It is therefore not possible to estimate the imaginary part independently of the real part.

In general, measured RIs of light-absorbing carbonaceous substances show a high variability caused by different burning conditions (Bond and Bergstrom, 2006). In our opinion, charcoal particles should share some of the radiative properties of coal

with similar H-C and O-C ratios (i.e. very low-ranked coal). If we slightly extend the "coal rank" line in Fig. 7 from Bond and Bergstrom (2006) to the H-C and O-C ratios of charcoal, we arrive at a refractive index of $\mathrm{RI} \approx 1.75 - 0.1k$ (for a wavelength of $550\,\mathrm{nm}$). Unfortunately, this approach does not give us any information about the wavelength dependence of RI. However, the imaginary part of RI should not matter as much as the particle size: the charcoal particles are large compared with the dominant wavelengths of sunlight. If the absorption is relatively high, it is expected that no light penetrates to the interior of

the particle, so that only the "skin" of the particle absorbs and all light encountering the particle skin is attenuated (Tami Bond, personal communication). The aerosol absorption in our model scales with the aerosol mass and does therefore not account for this. Hence we would overestimate the absorption by charcoal when using $\mathrm{RI} \approx 1.75 - 0.1k$. When we conducted 5-year test simulations with different RI for charcoal (once using RI from BC, once RI from dust), we did not detect clear differences in the atmospheric lifetime of charcoal between the model simulations. The RI of charcoal might therefore not be too important

for its atmospheric transport. In the end, we decided to use the same RI as for dust ($\mathrm{RI} \approx 1.52 - 1.1 \times 10^{-3}k$); the lower absorption component of dust compared to charcoal should counteract that only part of the charcoal mass is expected to absorb radiation.

## Appendix C: Gravitational settling

To ensure numerical stability in aerosol gravitational settling, aerosol particles can only cross one model layer within one

timestep. However, only large particles close to the surface, where the model layers are thin, are affected. As an example, in the second layer near the surface, particles with $r_g > 56\,\mu\mathrm{m}$ are affected assuming Stokes velocity (with a particle density of $0.6\,\mathrm{g\,cm^{-3}}$). We expect that this velocity restriction might delay gravitational settling for large particles by up to a few time steps near the surface. However, this should not considerably change the spatial gravitational settling pattern since particles are not transported far horizontally within this time (as the horizontal scale of the grid is much larger than the vertical scale

and horizontal wind velocities are rather low near the surface). Furthermore, only a small number of charcoal particles are sufficiently large to be impacted.

## Appendix D: Calculation of number fluxes above threshold radius

To compare the simulated charcoal fluxes with observations, it is essential that only simulated fluxes of particles with $D_{\mathrm{M}} > 10\,\mu\mathrm{m}$ are considered. We calculated the total number of charcoal particles above this threshold directly before and after the

30 calculation of the removal processes (gravitational settling, dry deposition, wet deposition). From the difference, we calculate the fluxes to the surface as illustrated in Fig. A2.




Since the observations only consider pure charcoal particles, we should only take the charcoal component of the soluble giant mode into account when comparing to observations. Therefore, knowledge about the size distribution of the charcoal component is important. Our model assumes an ideal internal aerosol mixture, i.e. the total number of particles for the soluble giant mode is also representative for the number of charcoal particles in the soluble giant mode. The charcoal mass on the other

hand is only a fraction of the total soluble giant mode mass (e.g. sulphate in addition). Hence, the number size distribution of the charcoal particles in the soluble giant mode is shifted to smaller radii compared to the total soluble giant mode but also follows a log-normal distribution with the same $\sigma$ as the total soluble giant mode (see Fig. A3).

Due to a small inconsistency in the code, small negative surface fluxes can occur: the radius used to calculate the removal processes is only updated once per timestep, while the number and mass tracer tendencies are updated inbetween. Since we use

the tracer tendencies to calculate the radius before and after the removal process, our diagnostics do not use exactly the same radius as the calculations for the removal processes do. However, the error is negligible compared to the mean surface fluxes.

*Acknowledgements.* We thank the Swiss National Science Foundation (SNF) for granting the Sinergia project "Paleo fires from high-alpine ice cores", which funded this research and enabled the collaboration between different research disciplines (CRSII2_154450). This work was also supported by a grant from the Swiss National Supercomputing Centre (CSCS) under project ID s652. In addition, we are grateful to

Michael Sigl and Dimitri Osmont from the Paul Scherrer Institute for helping with the ice core preparation. Furthermore, we acknowledge the help of W. O. van der Knaap from the University of Bern, who extracted the raw data from ALPADAPA, and Peter Kunes, who counted the charcoal particles from Singhe Dzong. We are also thankful to Thomas Blunier from the Institute for Ice and Climate in Kopenhagen; he kindly provided us ice from the Summit ice core. We generally thank the developers of ECHAM-HAM(MOZ), who continuously improve the model. The ECHAM-HAMMOZ model is developed by a consortium composed of ETH Zürich, Max Planck Institut für Meteorologie,

Forschungszentrum Jülich, University of Oxford, the Finnish Meteorological Institute, and the Leibniz Institute for Tropospheric Research, and managed by the Center for Climate Systems Modeling (C2SM) at ETH Zürich.

*Competing interests.* All authors declare that no competing interests exist.



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





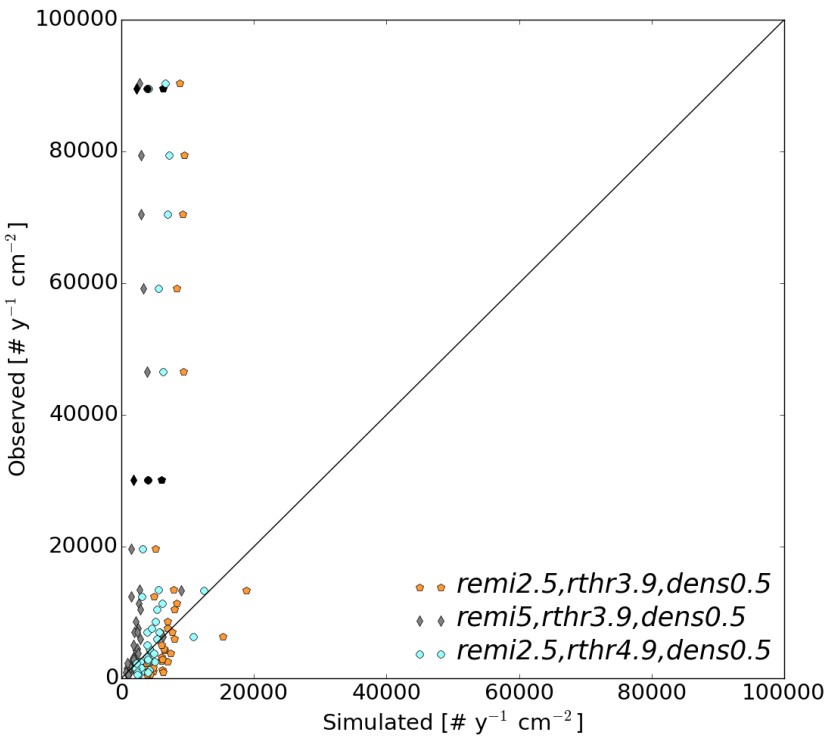

**Figure 1.** Observed versus simulated number fluxes of charcoal particles above the threshold radius (in $\mathrm{cm}^{-2}\,\mathrm{y}^{-1}$). Test simulations are shown for different combinations of emission number geometric mean radius ($remi$ in μm), threshold radius ($rthr$ in μm), and density ($dens$ in $\mathrm{g\,cm}^{-3}$). The scaling factors are identical ($SF = 34$). The black dots represent the two observation sites which are expected to be influenced by surface runoff (see Sect. 2.1).




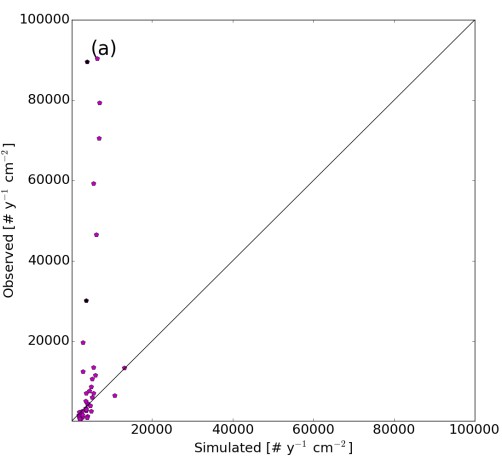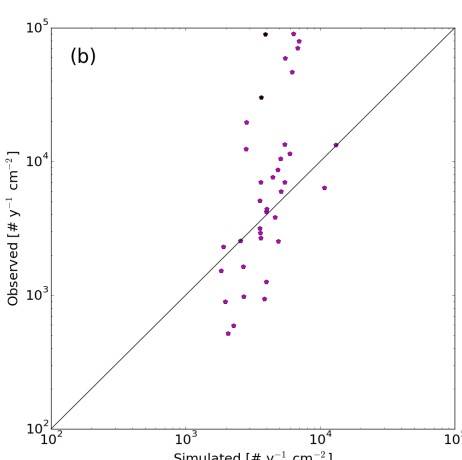

**Figure 2.** Simulated versus observed number fluxes of charcoal particles above the threshold radius (in $\mathrm{cm^{-2}\,y^{-1}}$) (a) on a linear scale and (b) on a logarithmic scale. The following parameters were used in this simulation: an emission number geometric mean radius of $r_{\mathrm{eq}} = 2.5\,\mu\mathrm{m}$, a threshold radius of $r_{\mathrm{eq}} = 4.9\,\mu\mathrm{m}$, and a charcoal density of $0.6\,\mathrm{g\,cm^{-3}}$. The scaling factor is $SF = 40$.





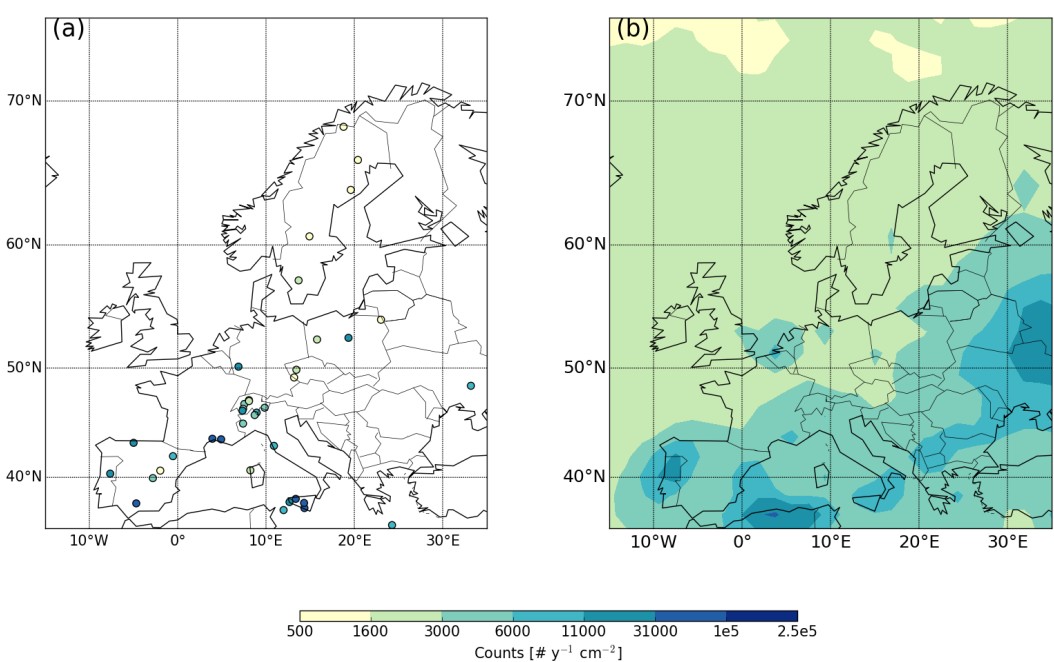

**Figure 3.** (a) Observed versus (b) simulated number fluxes of charcoal particles above the threshold radius (in $cm^{-2}\,y^{-1}$) using the best estimate of parameters (see caption of Fig. 2).





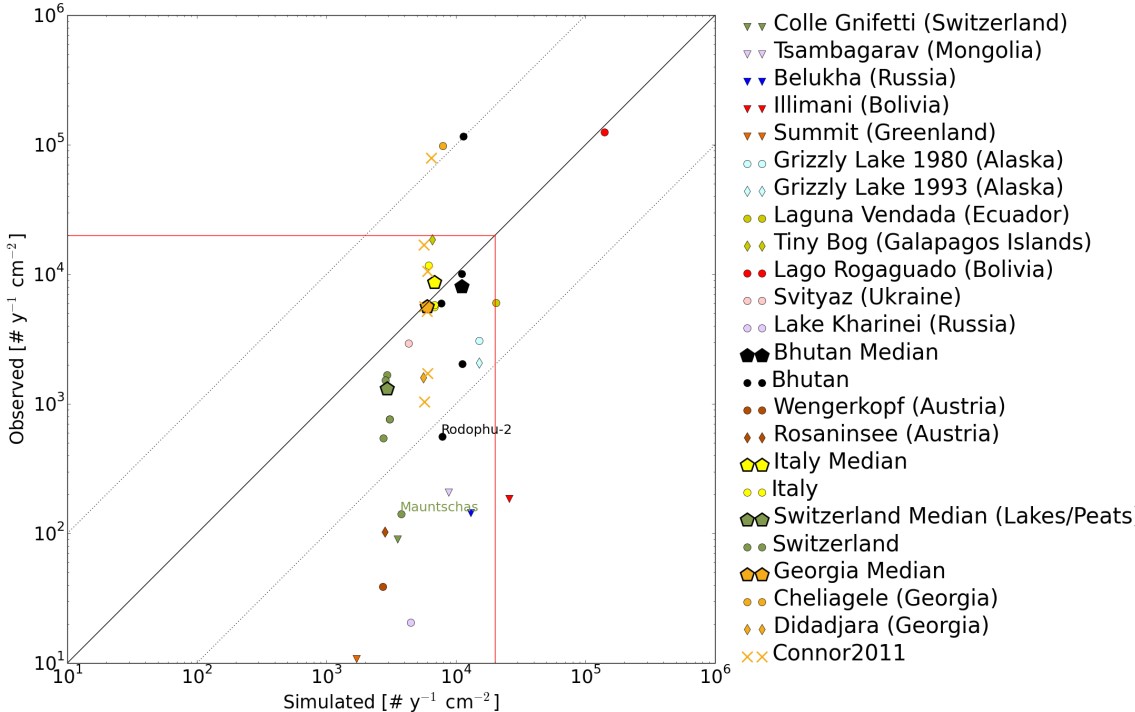

**Figure 4.** Simulated versus observed number fluxes of charcoal particles above a threshold radius of $r_{eq} = 4.9\,\mu$m (in $\text{cm}^{-2}\,\text{y}^{-1}$) for the validation dataset. The triangles refer to observations from ice cores; all other data is from sediments. The same colors are used for samples from the same countries. The data from Connor (2011) is distinguished by symbols (x-crosses) because a different method was used. To improve readability, the different sediment observations from Bhutan, Italy, Switzerland, and Georgia are not further distinguished in the legend since the observation sites in these countries are close together. The median over them is illustrated by the large pentagrams. The red frame shows the axis limits of Fig. 5. The black solid line is the 1:1 line; the lines that are one order of magnitude away from the 1:1 line are dotted.





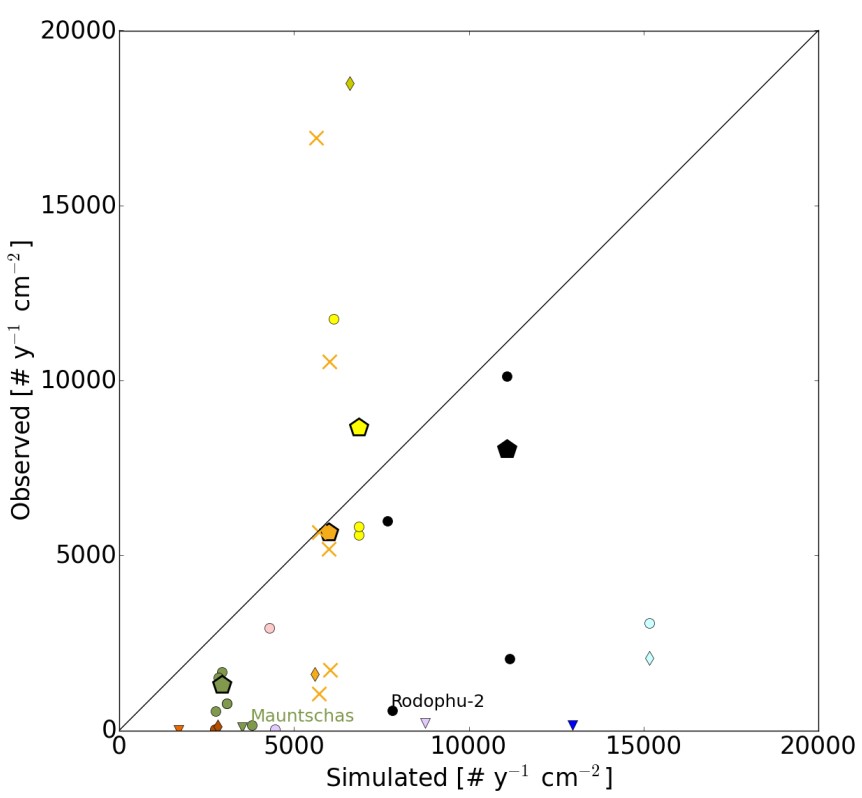

**Figure 5.** The same as Fig. 4 but on a linear scale with different axis limits (corresponding to the red frame in Fig. 4).



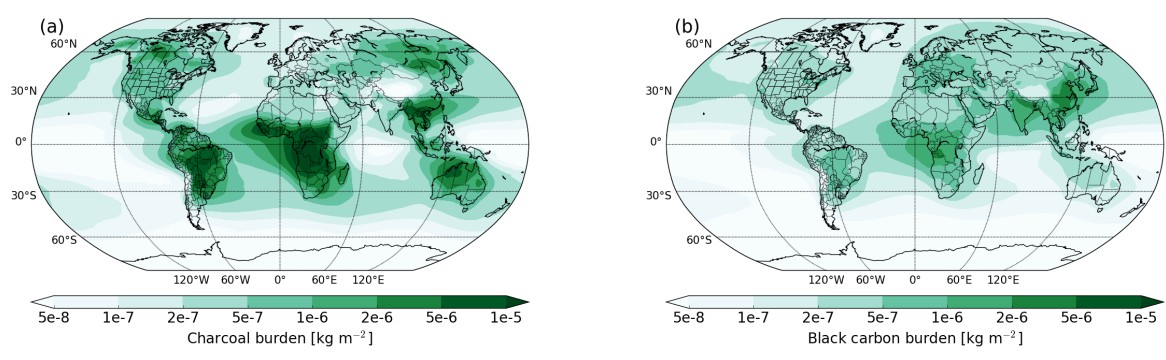

**Figure 6.** Simulated aerosol burden (all particle sizes) averaged over 10 years for (a) charcoal and (b) black carbon.





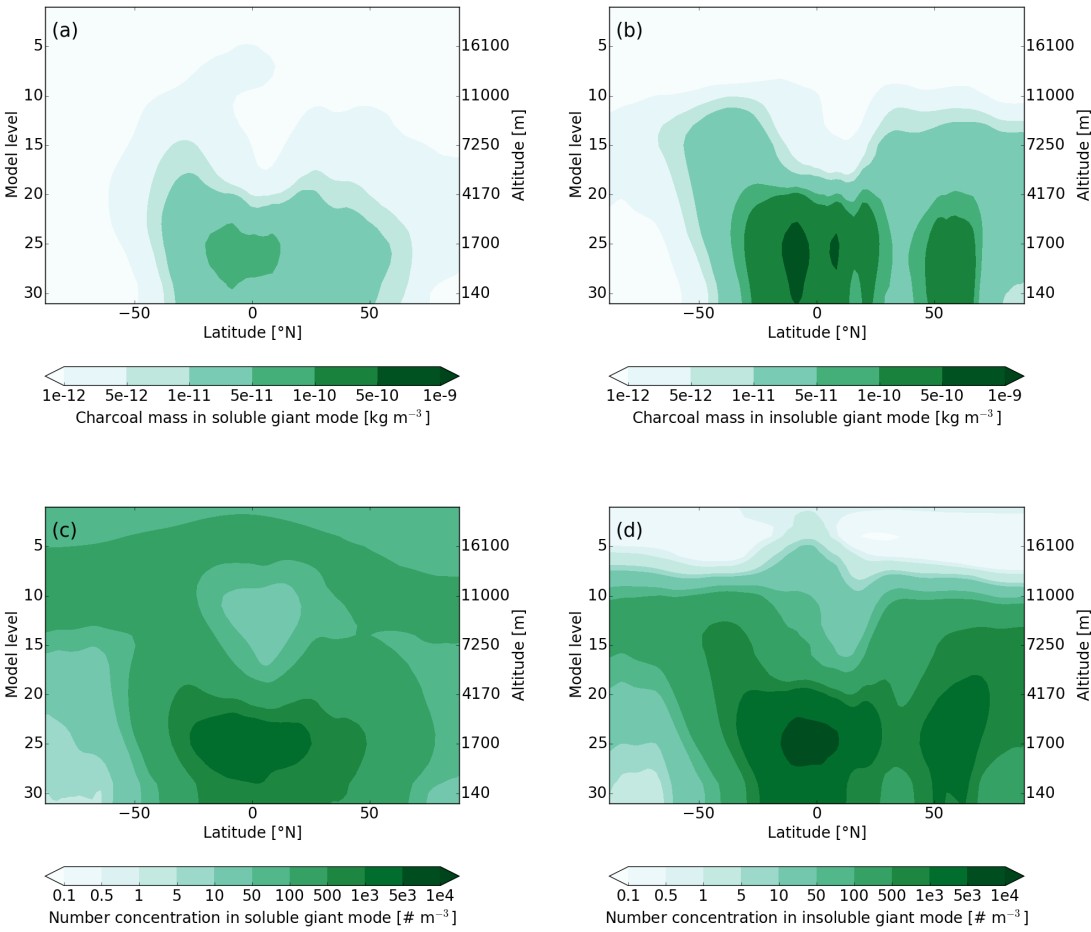

**Figure 7.** Ten-year zonal average of the charcoal mass concentration in (a) the soluble and (b) the insoluble giant mode and of the number concentration of (charcoal) particles in (c) the soluble mode and (d) the insoluble giant mode. The number geometric mean radius of the emitted particles is $2.5\,\mu m$. The right y-axis shows to which altitude the model layers approximately correspond to.





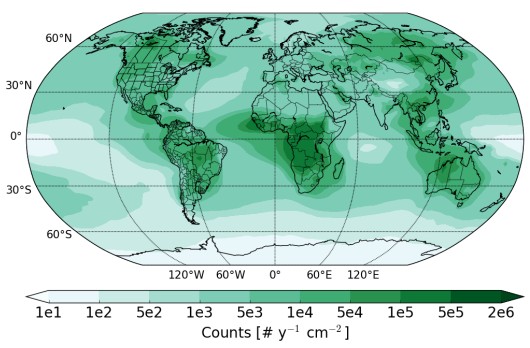

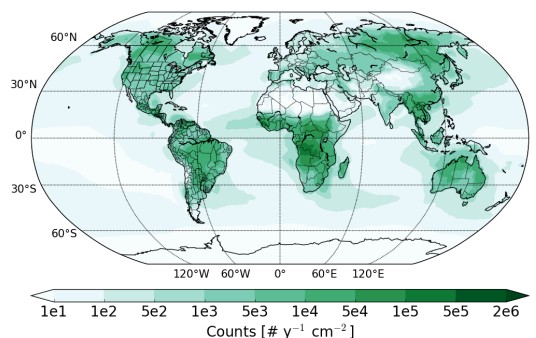

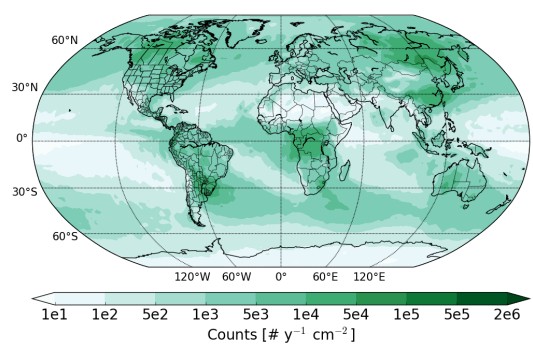

**Figure 8.** The number fluxes (number of particles above a threshold radius of $r_\text{eq} = 4.9\,\mu\text{m}$ in $\text{cm}^{-2}\,\text{y}^{-1}$) of the three different charcoal removal processes in the model: (a) gravitational settling, (b) dry deposition, and (c) wet deposition.





**Table 1.** Examplary results of test simulations with different parameters (emission number geometric mean radius $remi$ in µm, threshold radius $rthr$ in µm, and density $dens$ in $g\,cm^{-3}$). The scaling factor is the same for all simulations ($SF = 34$); the numbers hardly depend on the scaling factor. The parameters chosen for further simulations are marked in red.

| Parameters | Pearson correlation | Spearmean rank correlation | Quartile coefficient of dispersion |
|---|---|---|---|
| *remi2.5,rthr3.9,dens0.5* | 0.27 | 0.67 | 0.22 |
| *remi2.5,rthr3.9,dens0.6* | 0.27 | 0.67 | 0.24 |
| *remi2.5,rthr4.9,dens0.5* | 0.32 | 0.67 | 0.27 |
| *remi2.5,rthr4.9,dens0.6* | 0.32 | 0.68 | 0.30 |
| *remi5,rthr3.9,dens0.5* | 0.21 | 0.68 | 0.28 |
| *remi5,rthr4.9,dens0.6* | 0.23 | 0.68 | 0.38 |





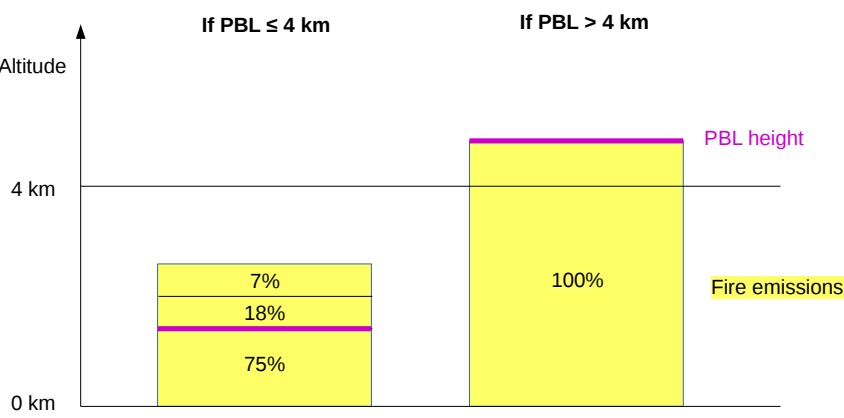

**Figure A1.** Illustration of fire emission heights in ECHAM6-HAM2.





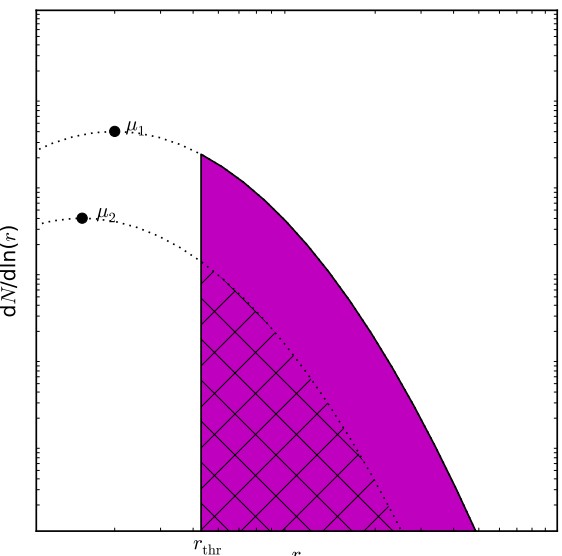

**Figure A2.** Illustration of how the deposition of charcoal particles above a certain threshold radius ($r_{\mathrm{thr}}$) was calculated. Before the removal process (e.g. gravitational settling), the number geometric mean radius of a gridbox is $\mu_1$. The number concentration of particles above the threshold radius is proportional to the area below the curve, i.e. the magenta area. After the removal process, both the number geometric mean radius and the total number concentration change (shift to $\mu_2$). Now the hatched area represents the particle number concentration above the threshold radius. From the difference between the magenta and the hatched area we can calculate how many charcoal particles are removed. Number fluxes are then calculated by dividing by the time step and the area of the gridbox.





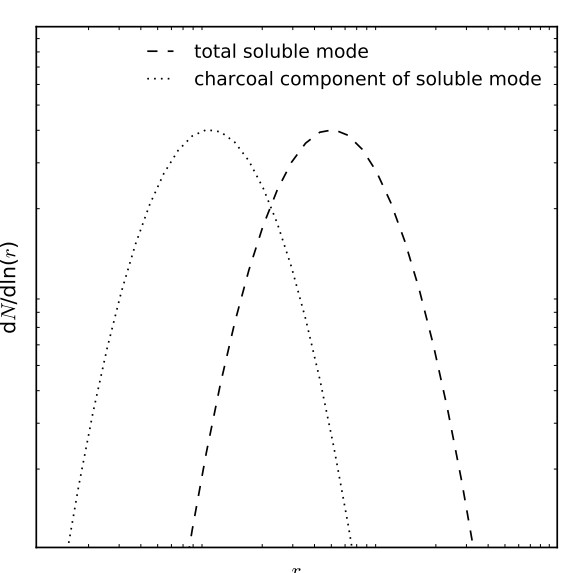

**Figure A3.** Schematic number size distribution of the total soluble giant mode and its charcoal component.





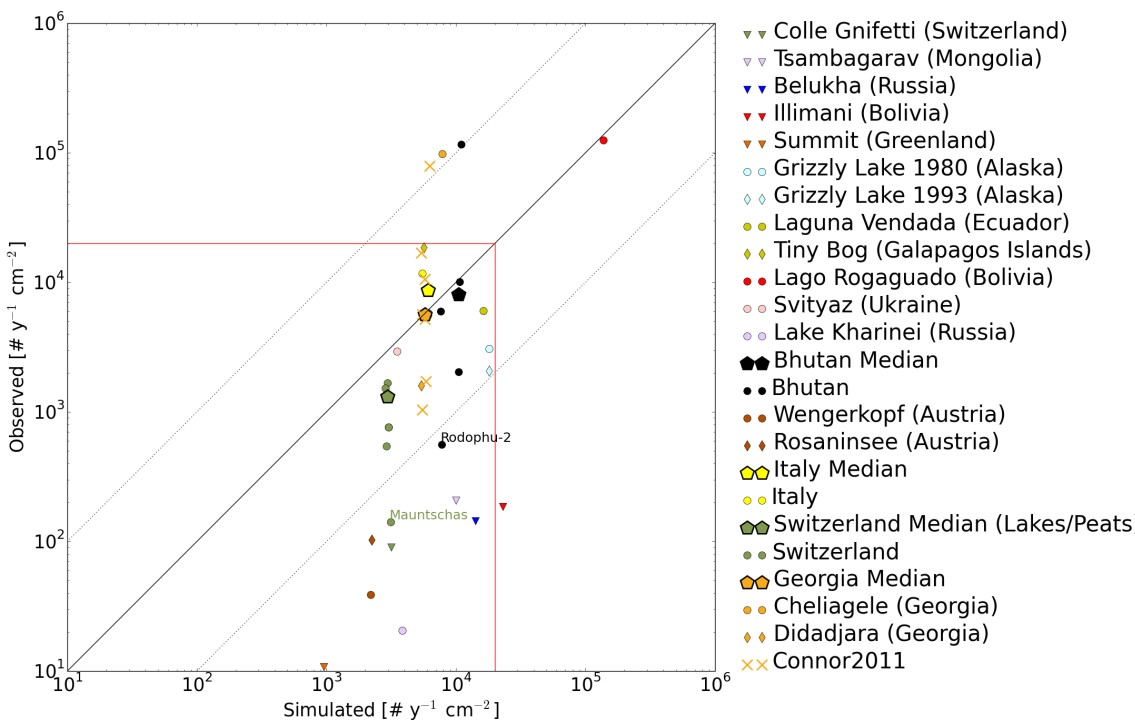

**Figure A4.** The same as Fig. 4 but for the free instead of the nudged model simulation.





**Table A1.** An overview of the observation sites from the calibration dataset (Adolf et al., 2017). The data is sorted alphabetically by country.

| Site | Country | Longitude [°] | Latitude [°] | Altitude [m ASL] | Lake size [ha] |
|------|---------|---------------|--------------|------------------|----------------|
| Černé jezero | Czech Republic | 13.18 | 49.18 | 1007 | 18.4 |
| Hromnické jezírko | Czech Republic | 13.44 | 49.85 | 332 | 2 |
| Étang d'Entressen | France | 4.92 | 43.6 | 34 | 103.5 |
| Lac du Crès | France | 3.93 | 43.65 | 39 | 6 |
| Holzmaar | Germany | 6.88 | 50.12 | 422 | 20 |
| Limni Kournas | Greece | 24.27 | 35.33 | 18 | 42.0 |
| Biviere di Gela | Italy | 14.35 | 37.02 | 0 | 150 |
| Gorgo Basso | Italy | 12.66 | 37.61 | 27 | 3 |
| Lago dell'Accesa | Italy | 10.9 | 42.99 | 111 | 16 |
| Lago dello Scanzano | Italy | 13.37 | 37.92 | 547 | 97 |
| Lago di Baratz | Italy | 8.23 | 40.68 | 24 | 60 |
| Lago di Pergusa | Italy | 14.31 | 37.52 | 677 | 50 |
| Lago di Varese | Italy | 8.72 | 45.83 | 240 | 1480 |
| Lago Piccolo d'Avigliana | Italy | 7.4 | 45.05 | 288 | 61.1 |
| Specchio di Venere | Italy | 11.99 | 36.82 | 8 | 19.4 |
| Jezioro Gołyń | Poland | 15.78 | 52.44 | 26 | 9.5 |
| Jezioro Gościąż | Poland | 19.34 | 52.58 | 76 | 42 |
| Suchar II | Poland | 23.02 | 54.09 | 140 | 2.5 |
| Lagoa Escura | Portugal | -7.64 | 40.36 | 1679 | 2 |
| Lago Enol | Spain | -4.99 | 43.27 | 1077 | 12.2 |
| Laguna Conceja | Spain | -2.81 | 39.93 | 857 | 29.4 |
| Laguna de Taravilla | Spain | -1.97 | 40.65 | 1113 | 2.1 |
| Laguna Grande de Estaña | Spain | -0.53 | 42.02 | 669 | 18.8 |
| Laguna Zóñar | Spain | -4.69 | 37.48 | 301 | 37 |
| Hagsjön | Sweden | 13.69 | 57.26 | 170 | 22.2 |
| Sarsjön | Sweden | 19.6 | 64.04 | 274 | 7.78 |
| Sisstjärnen | Sweden | 14.92 | 60.65 | 216 | 9.6 |
| Stora Utterträsk | Sweden | 20.41 | 66.12 | 277 | 28.1 |
| Vuolep Njakajaure | Sweden | 18.78 | 68.34 | 408 | 30 |
| Gerzensee | Switzerland | 7.55 | 46.83 | 603 | 25.2 |
| Iffigsee | Switzerland | 7.41 | 46.39 | 2065 | 10 |
| Lac du Mont d'Orge | Switzerland | 7.34 | 46.23 | 595 | 3 |
| Lago d'Origlio | Switzerland | 8.94 | 46.05 | 423 | 8 |
| Lej da San Murezzan | Switzerland | 9.85 | 46.49 | 1773 | 78 |
| Mauensee | Switzerland | 8.07 | 47.17 | 500 | 51 |
| Soppensee | Switzerland | 8.08 | 47.09 | 593 | 24 |
| Blue Lake | Ukraine | 33.2 | 48.45 | 87 | 24.4 |





Table A2: A summary of the different observation sites used for validation. Note that the youngest date of the record is given as *calibrated* years for the $^{14}C$ method. Most data was taken from the Alpine Pollen Database of University of Bern (ALPADABA). The sorting follows the legend of Fig. 4.

| Site | Country | Lon [°] | Lat [°] | Altitude [m ASL] | Time period | Record type | Dating method | Dated material | Youngest date | Lake size [ha] | Publication |
|---|---|---|---|---|---|---|---|---|---|---|---|
| Colle Gnifetti | Switzerland | 7.88 | 45.93 | 4450 | 2002-2015 | Ice core | Annual layer counting | - | - | - | This study |
| Tsambagarav | Mongolia | 90.85 | 48.66 | 4130 | 1988-2009 | Ice core | Annual layer counting | - | - | - | This study |
| Belukha | Russia | 86.59 | 49.81 | 4062 | Mean of two samples (1987, 1996/97) | Ice core | Annual layer counting | - | - | - | Eichler et al. (2011) |
| Illimani | Bolivia | -67.78 | -16.65 | 6300 | 2008-2015 | Ice core | Annual layer counting | - | - | - | This study |
| Summit | Greenland | -38.46 | 72.58 | 3200 | 1989 core | Ice core | Surface probe | - | - | - | This study |
| Grizzly Lake | Alaska | -144.19 | 62.71 | 720 | 1980, summer 1993 | Lake | $^{210}Pb$ | Bulk | AD1992-1994 | 11 | Tinner et al. (2006b) |
| Laguna Vendada | Ecuador | -79.39 | -3.61 | 3640 | 2009 | Peat | $^{14}C$ | Terrestrial macrofossil | AD943 ± 35 | 2 | This study |



| Site | Country | Latitude | Longitude | Depth | Year | Archive | Dating | Sphagnum/Material | Age | Value | Reference |
|---|---|---|---|---|---|---|---|---|---|---|---|
| Tiny Bog | Galapagos Islands (Ecuador) | 0.64 | -90.33 | 819 | 2005 | Peat | $^{14}C$ | Sphagnum | More recent than AD1950 | 0.01 | This study |
| Lago Rogaguado | Bolivia | -13.02 | -65.93 | 125 | 2004 | Lake | $^{14}C$ | Terrestrial macro-fossil | AD1543 ±84 | 31500 | Brugger et al. (2016) |
| Svityaz | Ukraine | 51.50 | 23.84 | 157 | 2004 | Lake | $^{14}C$ | Terrestrial macro-fossil | AD1139 ±110 | 2519 | This study |
| Lake Kharinei | Russia | 67.37 | 62.75 | 110 | Summer 1993 | Lake | $^{210}Pb$ | Bulk | AD2003 | 5 | Salonen et al. (2011) |
| Tergang | Bhutan | 27.83 | 91.97 | 2260 | Summer 1992 | Peat | $^{14}C$ | Bulk | AD1097 | Very small | This study |
| Shamling | Bhutan | 27.77 | 91.12 | 2350 | 1990 | Lake/peat | $^{14}C$ | Terrestrial macro-fossil | AD1807 | - | This study |
| Rodophu-2 | Bhutan | 28.05 | 89.78 | 4530 | 2000 | Lake/peat | $^{14}C$ | Bulk | AD1189 | - | This study |
| Laya | Bhutan | 28.05 | 89.68 | 3830 | 2000 | Lake/peat | $^{14}C$ | Terrestrial macro-fossil | AD1723 | - | This study |
| Singhe Dzong | Bhutan | 27.97 | 92.32 | 3800 | 2000 | Peat | $^{14}C$ | Terrestrial macro-fossil | AD1248 | Very small | P. Kunes |
| Wengerkopf | Austria | 47.17 | 13.87 | 1780 | Summer 2001 | Peat | $^{210}Pb$ | Bulk | AD1996 | 0.3 | van der Knaap et al. (2012) |



| Site | Country | | | | | | | | | | Reference |
|---|---|---|---|---|---|---|---|---|---|---|---|
| Rosaninsee | Austria | 13.78 | 46.95 | 2070 | Summer 2001 | Peat | $^{210}$Pb | Bulk | AD1993 | 0.3 | van der Knaap et al. (2012) |
| Gorgo Basso | Italy | 12.65 | 37.62 | 6 | 2001 | Lake | $^{14}$C | Terrestrial macro-fossil | AD1772 | 3 | Tinner et al. (2009) |
| Gorgo Longo di Ficuzza | Italy | 13.41 | 37.90 | 877 | 2003 | Lake | $^{14}$C | Terrestrial macro-fossil | AD755 | 0.2 | This study |
| Gorgo Tondo di Ficuzza | Italy | 13.41 | 37.90 | 783 | 2006 | Lake | $^{14}$C | Terrestrial macro-fossil | AD1280 | 0.6 | This study |
| Etang de la Gruère | Switzerland | 7.05 | 47.24 | 1005 | 1993 | Peat | $^{14}$C | Terrestrial macro-fossil | AD1809 | 22.5 | Public data (counted by van Leeuwen) |
| Mauntschas | Switzerland | 9.85 | 46.49 | 1819 | 2003 | Peat | $^{14}$C | Sphagnum | AD1988 | 10 | van der Knaap et al. (2012) |
| Les Amburnex | Switzerland | 6.23 | 46.54 | 1375 | 2000 | Peat | $^{14}$C | Terrestrial macro-fossil | AD1770 | 0.2 | Sjögren and Lamentowicz (2008) |
| Sèche de Gimel | Switzerland | 6.23 | 46.54 | 1300 | 2003 | Peat | $^{14}$C | Terrestrial macro-fossil | AD1995 | 12 | Sjögren (2005) |



Atmospheric Chemistry and Physics Discussions — Open Access EGU

| Site | Country | | | | Year | Archive | Dating | Dated material | Basal date | | Reference |
|---|---|---|---|---|---|---|---|---|---|---|---|
| Le Moé | Switzerland | 6.22 | 46.54 | 1310 | 2002 | Peat | $^{14}C$ | - | AD1991 | 12 | Sjögren (2006) |
| Hallwilersee | Switzerland | 8.21 | 47.28 | 400 | Summer 1998 | Lake | Annual layer counting | Sphagnum and Polytrichum stems | - | 1030 | This study |
| Lac de Bretaye | Switzerland | 7.07 | 46.33 | 1780 | 2012 | Lake | $^{14}C$ | Terrestrial macrofossil | BC82 | 4 | Thöle et al. (2016) |
| Cheliagele | Georgia | 43.11 | 42.62 | 1100 | Summer 1992 | Lake/peat | $^{14}C$ | - | AD1949 | - | This study |
| Didadjara | Georgia | 42.5 | 41.67 | 1850 | 2000 | Lake/peat | $^{14}C$ | - | AD859 | - | Connor et al. (2017, submitted) |
| Sakhare | Georgia | 45.32 | 41.58 | 800 | 2000 | Lake | $^{14}C$ | Bulk | AD500 ± 80 | 12 | Connor (2011) |
| Kumisi | Georgia | 44.83 | 41.58 | 469 | 2000 | Lake | $^{14}C$ | Bulk | AD800 ± 40 | 120 | Connor (2011) |
| Tsavkisi | Georgia | 44.75 | 41.68 | 1110 | 2000 | Peat | $^{14}C$ | Isolated pollen | AD530 ± 60 | 4 | Connor (2011) |
| Imera | Georgia | 44.2 | 41.65 | 1610 | 2000 | Lake | $^{14}C$ | Bulk | AD940 ± 40 | 12 | Connor (2011) |
| Bareti | Georgia | 44.17 | 41.65 | 1630 | 1986 | Lake | $^{14}C$ | Bulk | AD1050 ± 40 | 62 | Connor (2011) |



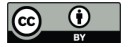

| | | | | | | | Correlation with neighbouring sites | ¹⁴C | | Connor (2011) |
|---|---|---|---|---|---|---|---|---|---|---|
| Jvari | Georgia | 44.73 | 41.83 | 570 | 2000 | Lake | - | - | 9.6 | Connor (2011) |
| Aligol | Georgia | 44.02 | 41.63 | 1534 | 2000 | Lake | Bulk | AD1760 ± 40 | 9.6 | Connor (2011) |