# Peer review of "Implementing Microscopic Charcoal Particles Into a Global Aerosol-Climate Model"

_Atmospheric Chemistry and Physics, 2017_

## Referee Comment (RC1) · BI Magi (Referee) · 28 Mar 2018

A very interesting modeling study to incorporate charcoal particles into the aerosol model using relatively modern charcoal accumulation rates. The work may prompt some thought in the Global Charcoal Database community about how charcoal gets transported in a full circulation model and what that may mean for interpreting charcoal accumulation rates relative to burned area. The authors describe the strengths and limits of the modeling approach nicely. I only have minor comments below that aim at clarifying a few points. Otherwise the results and technical implementation are quite useful.

p 2, line 10-11 Great statement, but the awkward sentence needs attention: "Open

questions still remain, e.g. regarding the complexity needed for global fire models (Hantson et al., 2016). Especially the anthropogenic influence on fires is difficult to simulate."

p. 2, line 12 Emissions are not calibrated exactly, but rather scaled to modern day. See Van Marle et al 2017 which is probably a paper that should be cited in this paragraph https://www.geosci-model-dev.net/10/3329/2017/

p. 3, lines 4-6 again, the Van Marle et al 2017 GMD paper would be a relevant citation here

p. 3, line 6 This makes it sound like Power and Marlon papers are "circumventing" the problem of comparing with global fire models. Re-word so that it is clear that they are circumventing problems associated with non-standardized data collection methods in the GCD.

p. 3, line 15 Useful citation here may be https://onlinelibrary.wiley.com/doi/full/10.1002/env.2450

p. 8 Radke et al: interesting use of the results from this study!

p. 8, lines 11-18 define the acronymn GFAS here; the text states that ratio of BC to submicron aerosol mass is 10 and that even BC emissions are likely underestimated by another factor of 3.4. Right now, this is confusing to me. I can see scaling up BC mass emissions as a starting point to simulate charcoal mass emissions, and I can see the extra factor of 3.4 or so arising from what may be an underestimate in BC mass emissions, but I cannot make sense of "are comparable to those of submicron particles and thus arrive at a factor of 10 based on the ratios of BC to total submicron particles and to OC". Please check the wording and clarify how SF = 34 is the starting point. Also, please clarify why SF = 40 is not used throughout the paper. For example, at line 18, why not add to the end of the last sentence "and we arrived at SF = 40 after an iterative calibration process."?

p. 9, line 19 change "which is like charcoal an inert and unreactive substance" to "which

is chemically similar to charcoal" (if this is what you are trying to say)

p. 10 section 3.1.4 the section seems overly speculative and distracting given the main goal of the paper. I agree that it might be interesting if micro and macro char were INPs but it seems equally as likely that if charcoal injected above 4 km in figure A1 is rare, then charcoal participating in Bergeron-W-F process is essentially insignificant in the model

p. 10, line 18-19 how could absorption of light leading to convective lifting of 5-10 micron particles? Can this tiny number of giant particles relative to submicron aerosol really have a dramatic impact on thermodynamic profile? Again, I find this distracting in the context of the main point of the paper, and would suggest simply stating that there is very limited study of the possibility of charcoal as INP. or CCN.

p. 11, lines 24-25 define what ACCMIP. stands for, and explain what "calculated online" means since the paper crosses across communities of researchers who may not guess what this model jargon mean

Conclusions: Several researchers in the Global Charcoal Database/Paleofire community published a study echoing some of the ideas in the conclusion that might be a useful citation supporting ideas around, line 10 on p. 15 https://www.sciencedirect.com/science/article/pii/S104061821630831X

Figures 1-5 in the caption, I suggest stating what the threshold radius actually is set to in each caption. Currently this is sometimes done and sometimes not.

Figure 3 is really interesting!

Figures 6-7 Section 3.1.6/Appendix D are a very useful diagnostic filter for subsetting model output to better match charcoal, but while Figs 1-5 and 8 show results filters for sizes greater than the threshold radius, this is not the case for Figure 6-7. Please speculate on roughly what fraction of the modeled burden might be due to the largest sized giant aerosols, and perhaps include this speculation as a part of the figure caption.

[Figure]

---

## Referee Comment (RC2) · Anonymous Referee #2 · 1 May 2018

The authors present the technical implementation of charcoal particles into a global climate model, calibrate emissions using a test dataset, and then evaluate their initial model performance against a global set of observations.

Understanding microscopic charcoal emissions, transport and deposition is clearly a relevant topic, both for atmospheric modeling and for climate studies, and the construction of models capable of dealing with this class of particle is very welcome. The paper is thorough and well written, and well suitable for publication in ACP. I do, however, have some comments on the evaluation the authors perform of their model, and the conclusions they draw. Also, as with many technical papers, quite a number of statements and sentences are difficult to understand for a broader audience. Hence, I recommend major revisions before final publication.

Major comments:

My main concern with the paper is about the emission scaling and validation against observations. The authors state very clearly that their initial implementation of charcoal particles fails to capture the full range of variability in the observations. This is quite understandable, and improving this correspondence should a fruitful and important line of research in coming years. Looking at Figures 1 and 2, however, it's clear that both the scaling factor used for emissions, and the parameters chosen for density and mean/threshold radius, don't really affect the correspondence much. I understand that the parameters are guided by observations (section 3.1.2), but both the early figures and Table 1 show that varying them don't really change the correlation.

I believe that the reason for this is that there are physical processes, both in transport and charcoal retrieval, that are not represented in the model - as the authors also comment on. Hence, the whole discussion of parameters and emission factors could be toned down quite a bit. It's relevant as a sensitivity test, and should be mentioned, but as the paper stands it seems to indicate that some conclusion about the "best" parameter set can be drawn - and I don't think the numbers support that.

Further, the emission scaling is confusing. The numbers 34 and 40 are used interchangeably through the manuscript - probably indicating that the lack of wide range correlation with observations precludes a more precise estimate. I would think that sensitivity study of emission scaling factors would be as, or more, important than the microscopic parameters discussed - so here, I would encourage the authors to add a little bit more information on how the scaling was chosen. (Especially since the abstract states that a factor of approx. 40 matches the calibration dataset "best".)

In conclusion, I recommend harmonizing the detail level in the discussions of microscopic parameters and emission factor, and admitting more clearly that the lack of variability in the model results precludes drawing firm conclusions about either. The implementation itself is important enough to warrant publication.
**Interactive comment**

Minor comments:

- Section 3.1.4: Interactions with radiation and clouds is a whole other topic, which is insufficiently covered by this section. E.g. the assumption about spherical particles will have large implications for radiative transfer. I recommend removing this discussion and taking it up more thoroughly in a later publication.

- p3,l1: Should be DM<10 micrometers for microscopic?

- p3,l7-8: "homogenised the variance of individual records with a Box-Cox transformation, rescaled the transformed data to the range (0, 1), and standardised it." This is a good example of a line that is too technical for its context. I recommend making the introduction more accessible to a broader audience.

- p6,l27: "The right-skewed histogram of Clark and Hussey"... is another example. Please explain, so the uninformed reader doesn't have to look though the references.

- p10,l1: Here, the authors describe an ageing process of charcoal particles, which likely influences wet deposition rates. What is the ageing timescale? How sensitive are the results (and the variability) to this parameter?

- p10,l21: "...nobody has measured..." Out of curiosity, could FireLab (https://www.firelab.org/) have performed some relevant experiments here? I saw something at a conference a while back, but can't quite recall the details.

- p11,l12: "...and some uncertain parameters..." Which ones? Please be specific. (It's described below, so it's just a matter of wording.)

- p11,l29: "Pearson correlation coefficients larger than 0.2." Is this the requirement for significance for your number of degrees of freedom? Please discuss the significance criteria in a bit more detail. (It's mentioned in a few places that the correlation becomes significant at the 5% level; according to what test? Which numbers are not significant?) This is related to my main comment above.

- p15,l25: Appendix B: Is there any way, based on the present data, to estimate the contribution of charcoal particles to the global absorption aerosol optical depth? Probably not with great precision, but this is a quite open issue. (See e.g. a recent review here: https://link.springer.com/article/10.1007%2Fs40641-018-0091-4)
* * *

---

## Author Response (AR1)

We thank Brian Magi for his helpful and valuable comments. Please note that we found a bug in the charcoal code, which affected the results. The following main points have changed:

- For the calibration data set, the Pearson correlations coefficients now show nearly no difference for different parameter sets (before: ranging between 0.21 and 0.32; now: ranging between 0.21 and 0.23).
- We chose a different parameter set with the new simulations, which has the highest variability (old parameter set: *remi2.5*, *rthr4.9*, *dens0.6*; new parameter set: *remi5*, *rthr4.9*, *dens0.6*).
- The emission factor that is in best agreement with the observations is now 250 (before: 40). This is both due to the error in the code and due to the different parameter set that we chose. We now discuss the effect of the parameter set on the estimated emission factor in the text.

Furthermore, based on the comments of reviewer 2, we decided to create a Supplementary Material, which now includes part of the manuscript and the Appendix. In the following, the reviewer's comments are shown in italic font and our answers in **bold** font.

A very interesting modeling study to incorporate charcoal particles into the aerosol model using relatively modern charcoal accumulation rates. The work may prompt some thought in the Global Charcoal Database community about how charcoal gets transported in a full circulation model and what that may mean for interpreting charcoal accumulation rates relative to burned area. The authors describe the strengths and limits of the modeling approach nicely. I only have minor comments below that aim at clarifying a few points. Otherwise the results and technical implementation are quite useful.

Thank you very much for your positive comments.

- p 2, line 10-11 Great statement, but the awkward sentence needs attention: "Open questions still remain, e.g. regarding the complexity needed for global fire models (Hantson et al., 2016). Especially the anthropogenic influence on fires is difficult to simulate."
- We agree and deleted the sentence "Especially the anthropogenic influence on fires is difficult to simulate.", which does not really fit in the context here.
- p. 2, line 12 Emissions are not calibrated exactly, but rather scaled to modern day. See Van Marle et al 2017 which is probably a paper that should be cited in this paragraph https://www.geosci-model-dev.net/10/3329/2017/
- The term "calibrated" is actually used in Rabin et al. (2017), but we agree that the term might be too strong (especially concerning aerosol emissions). We changed the sentence to: "Current fire models are generally tuned to match observations from recent decades, where satellite products give valuable information on the occurrence of fires."
- p. 3, lines 4-6 again, the Van Marle et al 2017 GMD paper would be a relevant citation here
- We agree that this paper should be cited in the introduction. We added the following text: "To estimate fire emissions from 1750 to 2015, van Marle et al. (2017) combined satellite retrievals, standardised scores from charcoal records, fire models, and visibility observations. The charcoal signal and the output from the fire models were scaled to match average regional GFED (Global Fire Emissions Database) carbon emissions from 1997 to 2003."

- p. 3, line 6 This makes it sound like Power and Marlon papers are "circumventing" the problem of comparing with global fire models. Re-word so that it is clear that they are circumventing problems associated with non-standardized data collection methods in the GCD.
- We changed the text to: "To circumvent the problem of inhomogeneous data..."
- p. 3, line 15 Useful citation here may be https://onlinelibrary.wiley.com/doi/full/10.1002/env.2450
- We added this citiation.
- p. 8 Radke et al: interesting use of the results from this study!
- p. 8, lines 11-18 define the acronymn GFAS here; the text states that ratio of BC to submicron aerosol mass is 10 and that even BC emissions are likely underestimated by another factor of 3.4. Right now, this is confusing to me. I can see scaling up BC mass emissions as a starting point to simulate charcoal mass emissions, and I can see the extra factor of 3.4 or so arising from what may be an underestimate in BC mass emissions, but I cannot make sense of "are comparable to those of submicron particles and thus arrive at a factor of 10 based on the ratios of BC to total submicron particles and to OC". Please check the wording and clarify how SF = 34 is the starting point. Also, please clarify why SF = 40 is not used throughout the paper. For example, at line 18, why not add to the end of the last sentence "and we arrived at SF = 40 after an iterative calibration process."?
- We changed the text to make it clearer: "As a starting point for the scaling factor, we assume that the mass emission fluxes of microscopic charcoal are comparable to those of submicron particles. Since BC only contributes relatively little to the total submicron particle mass, we scale the BC mass by a factor  $\approx 10$  (based on the ratios of BC to total submicron particles and to OC; Desservettaz et al. 2017, Akagi et al. 2011, Sinha et al. 2003). Furthermore, scaling aerosol emissions from the Global Fire Assimilation System (GFAS) by a factor of 3.4 leads to a better agreement between simulated and observed aerosol optical depth for both the global Monitoring Atmospheric Composition and Change (MACC) aerosol system and ECHAM6-HAM2 (Kaiser et al. 2012, Hardenberg et al. 2012). Therefore, we use a factor of  $10 \cdot 3.4 = 34$ as an initial estimate. Then we adjust this scaling factor until the simulated charcoal fluxes agree with the calibration dataset (Sect. 2.1)." Furthermore, as mentioned at the beginning, the best scaling factor is not 40 anymore. We now show in the supplementary material a plot with different scaling factors for the chosen parameter set.
- p. 9, line 19 change "which is like charcoal an inert and unreactive substance" to "which is chemically similar to charcoal" (if this is what you are trying to say)
- In our opinion, the term "chemically similar to charcoal" is too general since charcoal and black carbon differ considerably e.g. concerning their molecular composition. Therefore, we kept the words "inert" and "unreactive".
- p. 10 section 3.1.4 the section seems overly speculative and distracting given the main goal of the paper. I agree that it might be interesting if micro and macro char were INPs but it seems equally as likely that if charcoal injected above 4 km in figure A1 is rare, then charcoal participating in Bergeron-W-F process is essentially insignificant in the model
- We generally agree and shortened the section considerably.

- p. 10, line 18-19 how could absorption of light leading to convective lifting of 5-10 micron particles? Can this tiny number of giant particles relative to submicron aerosol really have a dramatic impact on thermodynamic profile? Again, I find this distracting in the context of the main point of the paper, and would suggest simply stating that there is very limited study of the possibility of charcoal as INP. or CCN.
- We shortened the section and added the following sentence to point out that the choices of refractive index and INP propensity have probabily not a large effect on our results: "We do not expect that these decisions have a large impact on the atmospheric transport of charcoal particles since most charcoal particles do not reach levels where heterogeneous freezing becomes important and the absorption of charcoal particles is likely too small to change the thermodynamic profile of the atmosphere."
- p. 11, lines 24-25 define what ACCMIP. stands for, and explain what "calculated online" means since the paper crosses across communities of researchers who may not guess what this model jargon mean
- We defined ACCMIP and changed the text to: "Dust, sea salt, and oceanic dimethyl sulphide emissions were calculated within the model at every timestep."
- Conclusions: Several researchers in the Global Charcoal Database/Paleofire community published a study echoing some of the ideas in the conclusion that might be a useful citation supporting ideas around, line 10 on p. 15 https://www.sciencedirect.com/science/article/pii/S1040618216
- We mention now this reference in the conclusions.
- Figures 1-5 in the caption, I suggest stating what the threshold radius actually is set to in each caption. Currently this is sometimes done and sometimes not.
- We added the used parameters to each caption.
- Figure 3 is really interesting!
- Figures 6-7 Section 3.1.6/Appendix D are a very useful diagnostic filter for subsetting model output to better match charcoal, but while Figs 1-5 and 8 show results filters for sizes greater than the threshold radius, this is not the case for Figure 6-7. Please speculate on roughly what fraction of the modeled burden might be due to the largest sized giant aerosols, and perhaps include this speculation as a part of the figure caption.
- We created additional diagnostics to calculate the contributions to the mass and the number online. Figures 6 and 7 (numbers in the old manuscript) now also show the mass/number of charcoal particles larger than the threshold radius so that they are directly comparable to the other figures.

We thank the reviewer for his helpful and valuable comments. Please note that we found a bug in the charcoal code, which affected the results. The following main points have changed:

- For the calibration data set, the Pearson correlations coefficients show now nearly no difference for different parameter sets (before: ranging between 0.21 and 0.32; now: ranging between 0.21 and 0.23).
- We chose a different parameter set with the new simulations, which has the highest variability (old parameter set: *remi2.5*, *rthr4.9*, *dens0.6*; new parameter set: *remi5*, *rthr4.9*, *dens0.6*).
- The emission factor that is in best agreement with the observations is now 250 (before: 40). This is both due to the error in the code and due to the different parameter set that we chose. We mention now in the text the effect of the parameter set on the estimated emission factor.

Furthermore, based on your comments, we decided to create a Supplementary Material, which now includes part of the manuscript and the Appendix. In the following, the reviewer's comments are shown in italic font and our answers in bold font.

The authors present the technical implementation of charcoal particles into a global climate model, calibrate emissions using a test dataset, and then evaluate their initial model performance against a global set of observations. Understanding microscopic charcoal emissions, transport and deposition is clearly a relevant topic, both for atmospheric modeling and for climate studies, and the con- struction of models capable of dealing with this class of particle is very welcome. The paper is thorough and well written, and well suitable for publication in ACP. I do, how- ever, have some comments on the evaluation the authors perform of their model, and the conclusions they draw. Also, as with many technical papers, quite a number of statements and sentences are difficult to understand for a broader audience. Hence, I recommend major revisions before final publication.

Major comments: My main concern with the paper is about the emission scaling and validation against observations. The authors state very clearly that their initial implementation of charcoal particles fails to capture the full range of variability in the observations. This is quite understandable, and improving this correspondence should a fruitful and important line of research in coming years. Looking at Figures 1 and 2, however, it's clear that both the scaling factor used for emissions, and the parameters chosen for density and mean/threshold radius, don't really affect the correspondence much. I understand that the parameters are guided by observations (section 3.1.2), but both the early figures and Table 1 show that varying them don't really change the correlation. I believe that the reason for this is that there are physical processes, both in transport and charcoal retrieval, that are not represented in the model - as the authors also comment on. Hence, the whole discussion of parameters and emission factors could be toned down quite a bit. It's relevant as a sensitivity test, and should be mentioned, but as the paper stands it seems to indicate that some conclusion about the "best" parameter set can be drawn - and I don't think the numbers support that. Further, the emission scaling is confusing. The numbers 34 and 40 are used inter- changeably through the manuscript - probably indicating that the lack of wide range correlation with observations precludes a more precise estimate. I would think that sensitivity study of emission scaling factors would be as, or more, important than the microscopic parameters discussed - so here, I would encourage the authors to add a little bit more information on how the scaling was chosen. (Especially since the abstract states that a factor of approx. 40 matches the calibration dataset "best".) In conclusion, I recommend harmonizing the detail level in the discussions of micro- scopic parameters and emission factor, and admitting more clearly that the lack of variability in the model results precludes drawing firm conclusions about either. The implementation itself is important enough to warrant publication. We perfectly agree with the reviewer. In principal, we cannot derive from our simulations which parameter set is the most realistic one because of the underestimated

variability and the weak Pearson correlation. Since the scaling factor depends on the parameter set, there is consequently also uncertainty concerning the scaling factor. We changed the following to account for the reviewer's comments:

- We changed the sentence in the abstract to: "We found that scaling black carbon fire emissions from the Global Fire Assimilation System (a satellitebased emission inventory) by approximately two orders of magnitude matches the calibration dataset best."
- We shortened the text about the realistic range of the parameters and partly moved it to the (new) supplementary material.
- The scaling factor of 34 was used as an initial estimate. The scaling factor was then adapted until it was in best agreement with the observations. In the old paper version, this was 40. We rewrote the Section "Calibration of emission": "We conducted test simulations and compared the result to the European observations from Adolf et al. (2018). Three measures were used for the comparison: i) the Pearson correlation, which is a measure for linear correlation; ii) the Spearman rank correlation, which assesses monotonic relationships; iii) the quartile coefficient of dispersion, which is a normalised and robust variability measure  $\left(\frac{Q_3-Q_1}{Q_3+Q_1}\right)$ , where  $Q_1$  and  $Q_3$  are the first and third quartiles, respectively). Table 1 shows some parameter combinations with positive correlation coefficients. In all test simulations, the correlation coefficients are very similar. While the Pearson correlation coefficients are low (0.21-0.23) and statistically insignificant, the Spearman rank correlation coefficients are much higher (0.67-0.69) and statistically significant. One reason for that are some observations with clearly larger charcoal fluxes than the simulated values ("outliers") because the Pearson correlation coefficients are much more sensitive to outliers than the Spearman rank correlation coefficients. These outliers can nicely be seen in Supplementary Fig. 4 for the example of remi2.5, rthr3.9, dens0.5." ... "The quartile coefficients of dispersion (Table 1) show that the variability differs between the test simulations. The simulation with the highest variability (remi5,rthr4.9,dens0.6; still having a lower variability than the observations, though) has only slightly lower correlation coefficients than the other simulations. Therefore, we choose this parameter set as the "best". However, we are aware that choosing the parameter set with the highest variability might compensate for errors not related to the parameters (e.g. the model resolution) that are responsible for an underestimated variability. Furthermore, none of the parameter sets has a statistically significant Pearson correlation. Therefore, we cannot conclude from our simulations which parameter set is the most realistic one.

For the chosen parameter set (*remi5*,*rthr4*.9,*dens0*.6), we conducted simulations with different scaling factors (see Supplementary Fig. 5). The correlation coefficients and the quartile coefficients of dispersion do hardly depend on the scaling factor because charcoal particles do not coagulate with each other. We did not use the root mean squared error as a measure for the best scaling factor because the charcoal observations span several orders of magnitudes and the absolute deviations would be biased by the highest absolute charcoal fluxes (including the outliers). Instead, we consider the scaling factor for which approximately the same number of observations lies above and below the 1:1 line to be in best accordance with the observations. This is the case for a scaling factor on the order of SF = 250 (see Supplementary Fig. 5c), which has furthermore the smallest mean absolute error. However, note that the scaling factor depends on the chosen parameter set. Considering all parameter sets listed in Table 1, the best scaling factors range between  $SF \approx 50$  and  $SF \approx 250$ ."

Minor comments:

- Section 3.1.4: Interactions with radiation and clouds is a whole other topic, which is insufficiently covered by this section. E.g. the assumption about spherical particles will have large implications for radiative transfer. I recommend removing this discussion and taking it up more thoroughly in a later publication.
- We shortened this section considerably and say now that we do not expect that the radiation and the interactions with cloud microphysics will have a large impact on our results.
- p3,l1: Should be DM<10 micrometers for microscopic?
- No, it is indeed  $D_M > 10 \ \mu m$  for microscopic and  $D_M > 100 \ \mu m$  for macroscopic charcoal.
- p3,17-8: "homogenised the variance of individual records with a Box-Cox transforma- tion, rescaled the transformed data to the range (0, 1), and standardised it." This is a good example of a line that is too technical for its context. I recommend making the introduction more accessible to a broader audience.
- We changed the text to: "To circumvent the problem of inhomogeneous data, global synthesis studies such as Power et al. (2008) and Marlon et al. (2008) homogenised, rescaled, and standardised the data."
- p6,127: "The right-skewed histogram of Clark and Hussey"... is another example. Please explain, so the uninformed reader doesn't have to look though the references.
- We deleted the word "right-skewed". It is not really necessary since the implication of the right-skewedness is mentioned in the text.
- p10,11: Here, the authors describe an ageing process of charcoal particles, which likely influences wet deposition rates. What is the ageing timescale? How sensitive are the results (and the variability) to this parameter?
- In ECHAM6-HAM2, no ageing timescale is calculated. The ageing i.e., coagulation and coating with sulphate is explicitly calculated while the model is running. We can therefore not easily adapt the ageing timescale. However, if the particles aged faster, then their lifetime would be lower. This is not only due to increased wet deposition, but also due to faster sedimentation, since the aged charcoal particles contain more material and are larger.
- p10,l21: "...nobody has measured..." Out of curiosity, could FireLab (https://www.firelab.org/) have performed some relevant experiments here? I saw something at a conference a while back, but can't quite recall the details.
- Thank you very much for this hint. Unfortunately, we could not find any measurement from firelab which covers the relevant size range for microscopic charcoal particles.
- p11,l12: "...and some uncertain parameters..." Which ones? Please be specific. (It's described below, so it's just a matter of wording.)
- We rewrote the text.

- p11,129: "Pearson correlation coefficients larger than 0.2." Is this the requirement for significance for your number of degrees of freedom? Please discuss the significance criteria in a bit more detail. (It's mentioned in a few places that the correlation becomes significant at the 5% level; according to what test? Which numbers are not significant?) This is related to my main comment above.
- As mentioned above, we rewrote the Section "Calibration of emissions" and make it now clear whether the results are statistically significant or not. Moreover, we added the following sentence to the methodology: "For comparing the simulations with the observations (e.g. calculating correlation coefficients), we used the SciPy package (Jones et al., 2001–)."
- p15,125: Appendix B: Is there any way, based on the present data, to estimate the contribution of charcoal particles to the global absorption aerosol optical depth? Prob- ably not with great precision, but this is a quite open issue. (See e.g. a recent review here: https://link.springer.com/article/10.1007%2Fs40641-018-0091-4)
- We added the following text to the Supplementary Material: "In our simulations, we found that the vertically integrated charcoal mass in the atmosphere is approximately one order of magnitude smaller than the mass of dust (using the chosen parameter set). Therefore, charcoal only contributes little to the total aerosol absorption optical thickness in our simulations. However, our simplified approach is very uncertain and does also not consider the nonsphericity of charcoal particles. If the absorption of charcoal were larger than with our simplified estimate, the contribution to the aerosol absorption optical thickness might be somewhat higher, although we do not expect it to be large."

The relevant changes include:

- Updated results after bug fix
- Shortening the section which describes how the charcoal parameters are estimated ( $r_{\rm emi}$ ,  $r_{\rm threshold}$ , density) and the section where the interactions of charcoal with clouds/radiation are discussed
- Highlighting the uncertainty of the chosen parameter set/scaling factor
- A better description of how the best scaling factor was determined
- Showing all charcoal results for particles larger than the threshold radius
- Using less technical terminology

**Implementing Microscopic Charcoal Particles Into a Global Aerosol-Climate Model**

Anina Gilgen1, Carole Adolf2,3,\*, Sandra O. Brugger2,3,\*, Luisa Ickes1,4, Margit Schwikowski5,3,6, Jacqueline F. N. van Leeuwen2,3, Willy Tinner2,3,7, and Ulrike Lohmann1

[revised manuscript text omitted]
 < rg < 500 nm), insoluble accumulation mode (50 nm < rg < 500 nm), insoluble accumulation mode, soluble coarse</li>
mode (500 nm < rg), and insoluble coarse mode. Each of these modes is log-normally distributed, and the total aerosol particle size distribution is described by a superposition of the seven modes. To implement charcoal particles, we extended the scheme by two additional modes (M9 scheme), namely by a soluble giant and an insoluble giant mode. We restricted neither the upper nor the lower bound of the giant mode but the rg of the emitted (i.e. initial) size distributions was set between 0.5 and 5 µm (see Sect. 3.1.2). When a particle size distribution grows in M7, part of its mass and number is shifted to the next larger mode, e.g. from the nucleation to the Aitken mode. To simplify diagnostics, we did not allow shifts from the coarse to the giant mode.

In HAM, all aerosol particles are assumed to be spherical. This condition is not fulfilled for charcoal particles but at least microscopic charcoal particles seem to have a shape closer to a sphere than macroscopic charcoal particles (Crawford and Belcher, 2014). To compare our result with observations, we therefore use the volume-equivalent radius (req) of charcoal particles. To estimate req, the geometry of charcoal particles must be considered. Some studies analysed the shape of charcoal particles and reported their aspect ratios R = DM/Dm, where Dm is the minimum dimension of a particle. We briefly In the

Supplementary Material (Sect. 1.1), we summarise the findings concerning R in literaturebefore explaining which range of R we consider in . In our model simulations.

The right-skewed histogram by Clark and Hussey (1996) shows a distinct maximum in the bin R = 1.5-2, and the mean aspect ratio is  $2.36 \pm 1.53$ . While Clark and Hussey (1996) used 9 sites in temperate eastern North America for their analysis,

- 5 Tinner and Hu (2003) studied charcoal particles from different biomes, namely Lago di Origlio (Switzerland; warm-temperate chestnut forest), Grizzly Lake (Alaska; spruce forest), and Wien Lake (Alaska; shrub birch tundra, poplar forest, and boreal forest). For the three sites, they report aspect ratios of R = 1.9, R = 1.7, and R = 1.6, respectively. Crawford and Belcher (2014) measured the aspect ratios of both microscopic and macroscopic charcoal particles. For microscopic charcoal ( $D_M$  up to 100), they found aspect ratios of 1.8-we consider a range of R between 1.33 and 2.4 for charcoal from wood and grass, respectively.
- 10 It is worth mentioning that they used a cross-sectional area of 315 as the lower threshold, which corresponds to a  $D_M$  of about  $11.5 13.4 \,\mu\text{m}$  for wood and  $13.2 15.5 \,\mu\text{m}$  for grass (assuming rectangual/elliptical cross-sections), i.e. a slightly larger  $D_M$  than the threshold of used in this study.

All of these measurements of R lie in the same range. For our study, we chose R = 2 as an initial estimate. The third, non-visible dimension of the particle is expected to be smaller or equal to  $D_{\rm m}$  for particles detected in pollen slides since

15 the particles may tend to lie flat on the slides (Clark and Hussey, 1996). For simplicity, we describe the shape of the charcoal particles with a rectangual cuboid (see Clark and Hussey, 1996). Assuming that the non-visible axis equals the minor axis  $D_{\rm m}$  (which is rather an upper estimate), the equivalent-volume radius  $r_{\rm eq}$  is given by:

$$\frac{V_{\text{cuboid}}}{\underline{D}_{\text{M}} \cdot \frac{D_{\text{M}}}{R} \cdot \frac{D_{\text{M}}}{R}} = \frac{4}{3} \cdot \pi \cdot r_{\text{eq}}^{3}}$$
$$\underline{\rightarrow r_{\text{eq}}} = \frac{4}{3} \cdot \pi \cdot r_{\text{eq}}^{3}}$$

20

where V stands for volume. The typical lower threshold for microscopic charcoal particles is  $D_{\rm M} = 10$ , which corresponds to an equivalent-volume radius of  $r_{\rm eq} \approx 3.9$ . However, since the aspect ratio tends to increase with charcoal size (Crawford and Belcher, 2014) , R of the lower threshold ( $D_{\rm m} = 10$ ) might be smaller than the mean or median R for  $D_{\rm m} > 10$ . In the model, we cannot account for a size-dependent R. For this study, it is important that the lower threshold of the counted and simulated charcoal

25 particles match well since these small particles have higher number concentrations than larger particles (Clark and Hussey, 1996; Tinner et al., As a lower estimate for our test simulations (see Sect. 3.2), we therefore use R = 1.33, which corresponds to the often applied, observation-based threshold of 75 for microscopic charcoal cross-sections (e.g. Tinner et al., 2006b) and which results in  $r_{eq} = 4.9$ . Based on the before mentioned observations from Clark and Hussey (1996) and Crawford and Belcher (2014), R = 2.4 is considered to be an upper bound. (corresponding to  $r_{eq}$  of  $4.9 \,\mu\text{m}$  and  $3.5 \,\mu\text{m}$ ); our initial estimate is R = 2

**30 (corresponding to $r_{\rm eq} = 3.9 \,\mu{\rm m}$ ).**

A distinct characteristic of charcoal particles is their low density. Renfrew (1973) reports values of  $0.3-0.6 \,\mathrm{g \, cm^{-3}}$ , Sander and Gee (1990) similar values of  $0.45-0.75 \,\mathrm{g \, cm^{-3}}$ . Hence, we chose a particle density of  $0.5 \,\mathrm{g \, cm^{-3}}$  as an initial guess, which

lies in the middle of these ranges. For the test simulations, we considered values where both observations overlap, i.e. from 0.45 to  $0.6 \,\mathrm{g \, cm^{-3}}$ .

**3.1.2 Charcoal emissions**

Thanks to fire emission inventories based on satellite data, we have a good knowledge about where and when fires of which

- 5 sizes occurred in the last 1-2 decades. Nevertheless, aerosol emissions from fires are still uncertain. This is caused to a large degree by the pronounced variability of fires: emission factors (which relate the mass of the burnt vegetation to the mass of emitted aerosol particles) vary considerably depending for instance on vegetation type, fire temperature, or fire dynamics. To our knowledge, no study has estimated the emission factors of microscopic charcoal particles so far. Clark et al. (1998) and Lynch et al. (2004) focused on macroscopic charcoal when estimating mass emission fluxes, therefore these values are not comparable.
  - Airborne measurements of aerosol particles from fires usually have upper cutoff sizes of a few micrometres or less (e.g. Johnson et al., 2008; May et al., 2014). The aircraft measurements by Radke et al. (1990) are exceptional since they include particles with sizes up to 3 mm, therefore covering the whole size range of charcoal. In their study, they set three fires in North America. The measured particle size distribution showed similar shapes for all of these burns. Radke et al. (1990) report that a
- 15 considerable fraction of the particles measured in the plumes were larger than 45 µm in diameter. From their data, we estimate that the mass emission fluxes of microscopic charcoal supermicron particles should be on the same order of magnitude as the mass emission fluxes of submicron particles, which is usually dominated by organic carbon (OC) in fire plumes (Desservettaz et al., 2017). By that we assume that all of these large particles are indeed charcoal and not ash or other large particles emitted from fires.
- Since both BC and charcoal particles form under conditions when oxygen is limited in the burning process, we decided to scale BC mass emissions from fires to derive charcoal mass emissions. As a starting point for the scaling factor, we assume that the mass emission fluxes of microscopic charcoal are comparable to those of submicron particles and thus arrive at a factor of . Since BC only contributes relatively little to the total submicron particle mass, we scale the BC mass by a factor  $\approx 10$  based on the ratios of BC to total submicron particles and to OC (Desservettaz et al., 2017; Akagi et al., 2011; Sinha et al., 2003). Note
- 25 that a larger factor of ≈ 34 might also be realistic since (based on the ratios of BC to total submicron particles and to OC; Desservettaz et al . Furthermore, scaling aerosol emissions from GFAS the Global Fire Assimilation System (GFAS) by a factor of 3.4 leads to a better agreement between simulated and observed aerosol optical depth for both the global Monitoring Atmospheric Composition and Change (MACC) aerosol system and ECHAM6-HAM2 (Kaiser et al., 2012; von Hardenberg et al., 2012). Therefore, we use a factor of 10 · 3.4 = 34 as an initial estimate. Then we adjust this scaling factor until the simulated charcoal fluxes
- 30 agree with the calibration dataset (Sect. 2.1).

To describe the fire emissions, we use BC, OC, and SO2 mass emissions at a 3-hourly resolution by combining the daily emissions from the Global Fire Assimilation System GFAS (GFASv1.0 until September 2014, GFASv1.2 afterwards) with the daily cycle from the Global Fire Emissions Database (GFED; Kaiser et al., 2012; Mu et al., 2011)GFED (year 2004; Kaiser et al., 2012; Mu et al., 2012; Mu et al., 2011)GFED (year 2004; Kaiser et al., 2012; Mu et al., 2012; Mu et al., 2011)GFED (year 2004; Kaiser et al., 2012; Mu et al., 2011)GFED (year 2004; Kaiser et al., 2012; Mu et al., 2011)GFED (year 2004; Kaiser et al., 2012; Mu et al., 2011)GFED (year 2004; Kaiser et al., 2012; Mu et al., 2011)GFED (year 2004; Kaiser et al., 2012; Mu et al., 2012) (year 2004; Kaiser et al., 2012; Mu et al., 2012) (year 2004; Kaiser et al., 2012; Mu et al., 2012) (year 2004; Kaiser et al., 2012; Mu et al., 2012) (year 2004; Kaiser et al., 2012; Mu et al., 2012) (year 2004; Kaiser et al., 2012; Mu et al., 2012) (year 2004; Kaiser et al., 2012; Mu et al., 2012) (year 2004; Kaiser et al., 2012; Mu et al., 2012) (year 2004; Kaiser et al., 2012; Mu et al., 2012) (year 2004; Kaiser et al., 2012; Mu et al., 2012) (year 2004; Kaiser et al., 2012; Mu et al., 2012) (year 2004; Kaiser et al., 2012; Mu et al., 2012) (year 2004; Kaiser et al., 2012; Mu et al., 2012) (year 2004; Kaiser et al., 2012; Mu et al., 2012) (year 2004; Kaiser et al., 2012; Mu et al., 2012) (year 2004; Kaiser et al., 2012; Mu et al., 2012) (year 2004; Kaiser et

dreae and Merlet (2001, with annual updates by M. O. Andreae). The strongest spurious signals originating from industrial activity, gas flaring, and volcanoes should be masked. However, in our simulations we found unrealistically high charcoal emissions over Iceland. These "emissions" are most likely caused by lava, which emits a signal at the same wavelength at which fires are detected. As an example, the volcano Bardarbunga caused huge eruptions over Iceland in August/September

- 5 2014, coinciding with extremely high fire emissions in GFAS  $(2.32 \cdot 10^{-11} \text{ kg m}^{-2} \text{ s}^{-1} \text{ averaged between } 62^{\circ} \text{ N } 26^{\circ} \text{ W}$  and  $67^{\circ} \text{ N } 11^{\circ} \
[revised manuscript text omitted]